# Charge density wave induced nodal lines in LaTe₃

**Shuvam Sarkar** [1], **Joydipto Bhattacharya**[2,3], **Pampa Sadhukhan**[1], **Davide Curcio**[4], **Rajeev Dutt** [2,3], **Vipin Kumar Singh** [1], **Marco Bianchi** [4], **Arnab Pariari**[5], **Shubhankar Roy**[6], **Prabhat Mandal**[5], **Tanmoy Das**[7], **Philip Hofmann** [4], **Aparna Chakrabarti**[2,3] & **Sudipta Roy Barman** [1] ✉

LaTe₃ is a non-centrosymmetric material with time reversal symmetry, where the charge density wave is hosted by the Te bilayers. Here, we show that LaTe₃ hosts a Kramers nodal line—a twofold degenerate nodal line connecting time reversal-invariant momenta. We use angle-resolved photoemission spectroscopy, density functional theory with an experimentally reported modulated structure, effective band structures calculated by band unfolding, and symmetry arguments to reveal the Kramers nodal line. Furthermore, calculations confirm that the nodal line imposes gapless crossings between the bilayer-split charge density wave-induced shadow bands and the main bands. In excellent agreement with the calculations, spectroscopic data confirm the presence of the Kramers nodal line and show that the crossings traverse the Fermi level. Furthermore, spinless nodal lines—completely gapped out by spin-orbit coupling—are formed by the linear crossings of the shadow and main bands with a high Fermi velocity.

Recent years have witnessed rapid development in the understanding of the physics of cooperative charge density wave (CDW) electronic state[1–12]. In particular, the interplay of the CDW electronic state with the non-trivial topological phases provides an interesting platform for the discovery of novel quasiparticles such as, axion insulator[3,13], quantum spin-Hall insulator[14], fractional Chern insulator states[15] and manipulation of topologically protected states[16,17]. CDW can drive topological phase transitions by modifying the symmetry of the lattice, such as breaking the inversion symmetry[18]. Interesting topological phases are frequently found in non-centrosymmetric materials, such as nodal chain fermions[19], Dirac and Weyl fermions[20–22], hourglass fermions protected by glide reflection[23], Kramers Weyl semimetal (KWS)[24] and recently predicted Kramers nodal line (KNL) metal[25]. KNLs differ from the Weyl nodal lines because they join two time reversal invariant momenta (TRIM) points and should appear in all achiral non-

centrosymmetric time reversal symmetry (TRS) preserving systems[25]. For the subclass of nonsymmorphic symmetry, KNLs emerge from the Γ points. Unlike the previously known nodal lines manifested by band inversion[26], the KNLs are robust under spin-orbit coupling (SOC) unless the protecting lattice symmetries such as TRS, mirror, or roto-inversion symmetries are broken. KNL fermions have been predicted to exhibit physical properties such as quantized optical conductivity[25]. However, in this emerging field, to the best of our knowledge, the experimental evidence of KNL is limited to the work by Shang et al.[27] who reported that transition metal ruthenium silicides belong to this class and exhibit unconventional superconductivity based on muon spin spectroscopy and density functional theory (DFT).

In recent years, multiple fascinating findings in LaTe₃[1,4,7,28–30]—a member of the RTe₃ (R represents a rare earth element) family with highest CDW transition temperature of 670 K[31,32]—have rekindled the

[1]UGC-DAE Consortium for Scientific Research, Khandwa Road, Indore 452001, Madhya Pradesh, India. [2]Theory and Simulations Laboratory, Raja Ramanna Centre for Advanced Technology, Indore 452013, Madhya Pradesh, India. [3]Homi Bhabha National Institute, Training School Complex, Anushakti Nagar, Mumbai 400094, Maharashtra, India. [4]Department of Physics and Astronomy, Interdisciplinary Nanoscience Center (iNANO), Aarhus University, Aarhus C 8000, Denmark. [5]Saha Institute of Nuclear Physics, HBNI, 1/AF Bidhannagar, Kolkata 700064, India. [6]Vidyasagar Metropolitan College, 39, Sankar Ghosh Lane, Kolkata 700006, India. [7]Department of Physics, Indian Institute of Science, Bangalore 560012, India. ✉e-mail: barmansr@gmail.com

scientific interest of the community in this TRS preserving material. The detection of an axial Higgs boson mode in LaTe$_3$ from Raman spectroscopy has been related to unconventional CDW excitation[1]. Photoinduced CDW state with topological defects in this material has been discovered from ultrafast electron diffraction and related studies[7,28]. The non-centrosymmetry in LaTe$_3$ was established from the structure determined by single crystal x-ray crystallography[33] as well as from the appearance of the $B_1$ symmetric Raman modes[34]. Transport studies have revealed that LaTe$_3$ possesses an unusually high non-saturated longitudinal magnetoresistance[30], which is similar to that of nodal line material[35]. In addition, LaTe$_3$ possesses very high carrier mobility[30], which, in conjunction with its high transition temperature, makes it a promising contender for next-generation electronics. Only a few angle resolved photoemission spectroscopy (ARPES) measurements on LaTe$_3$ have been reported in the literature[7,36]. Using time resolved ARPES, Zong et al.[7] examined light-induced melting of the CDW state of LaTe$_3$. A previous research by Brouet et al.[36] demonstrated that the CDW-induced shadow bands hybridize with the main bands existing in the non-CDW state creating a CDW gap along a particular direction of the Brillouin zone (BZ). Theoretical calculation of the electronic susceptibility as well as experiments have shown that $q$ dependent electron-phonon coupling plays an important role in stabilizing the CDW state in RTe$_3$[37-39]. A recent DFT study on a free-standing monolayer of LaTe$_3$ revealed that tensile strain would increase the CDW order, while compressive strain would suppress it, and superconductivity could develop[37]. In light of the fact that LaTe$_3$ is a non-centrosymmetric achiral material with TRS intact, topological phases brought on by inversion symmetry breaking in conjunction with other symmetries may be anticipated in the CDW state.

Here, from an in-depth study of the band structure of LaTe$_3$ single crystal in the CDW state by combining ARPES and ab−initio DFT using realistic experiment-based structure, we establish the existence of a KNL in a CDW material. It originates from the interaction of the shadow band and the main band, and is hosted by the TRS and the lattice symmetries. Furthermore, spinless nodal lines that are entirely gapped out by SOC are also identified.

## Results

### Modulated structure of LaTe$_3$ in the CDW state

The CDW in LaTe$_3$ has been reported to be unidirectional with an incommensurate wave vector $\mathbf{q}_{CDW}$ of $0.2757(4)\mathbf{c}^*$ from x-ray crystallography[33], where $\mathbf{c}^*$ is the reciprocal lattice vector along $k_z$ in the non-CDW state. It may be noted that an incommensurate structure can be represented as commensurate with a large unit cell such that its $\mathbf{q}_{CDW}$ is equal to the incommensurate value within the experimental accuracy of x-ray crystallography[40,41]. So, after considering different possible combinations of the numerator and the denominator of a fraction that could represent $\mathbf{q}_{CDW}$, we arrive at a 29-fold (29f) structure ($1 \times 1 \times 29$) derived from the experimental structure (see Methods) with $\mathbf{q}_{CDW} = \frac{8}{29}\mathbf{c}^* = 0.2759\mathbf{c}^*$ that matches experimental $0.2757(4)\mathbf{c}^*$ within its experimental accuracy. It has 232 atoms in the unit cell with positions almost coinciding with those given by x-ray crystallography [230 atoms have zero (0.0000 Å) displacement with respect to ref. 33, while only 2 atoms show a displacement of 0.0001 Å]. It should be mentioned that inorganic materials with such large unit cells do exist in nature[42-44]. The 29f structure has a non-centrosymmetric space group of $C2cm$ ($SG$ #40) that is the basic space group reported in ref. 33.

The 29f structure is unique because, although being commensurate, it represents the incommensurate structure of LaTe$_3$ within the experimental accuracy[40,41]. However, because of the large size of the unit cell, DFT calculations become computationally challenging and expensive. So, we consider a relatively smaller unit cell with 56 atoms (Fig. 1a) that has 7-fold (7f) modulated structure ($1 \times 1 \times 7$) with $\mathbf{q}_{CDW} = \frac{2}{7}\mathbf{c}^* = 0.2857\mathbf{c}^*$ with same symmetry as the 29f structure. Figure 1a shows that LaTe$_3$ is made up of two main structural units: the Te2-Te3 bilayer

that hosts the CDW and the La-Te1 corrugated slab. The Te bilayer, highlighted by blue double-sided arrows, is weakly coupled by van der Waals interaction. The primitive unit cell of the 29f structure is shown in Supplementary Fig. 1. Supplementary Fig. 2a, b shows the displacement of the Te atoms in the CDW state for both 7f and 29f structures with respect to the non-CDW positions. By comparing with the atom positions from the structure (.cif) file of ref. 33, we show that the atom positions are indistinguishable in the 29f structure. The 7f structure shows rather small deviations in both $\mathbf{q}_{CDW}$ (3.5%) and the amplitude of modulation (5%). In Supplementary Fig. 3, we show how the non-centrosymmetry arises in LaTe$_3$ by breaking the $M_x$ mirror symmetry. Also, the orientation of the polar axis along $x$ is indicated, which is dictated by the retained $M_z$ mirror and $\tilde{M}_y$ glide symmetries.

The signature of the CDW in the Te layer has been directly observed from our high resolution scanning tunneling microscopy (STM) topography image in Fig. 1b. The white solid lines show the distorted Te net formed by connecting the Te atoms (orange circles). Also, the white dashed lines showing the average positions of the adjacent Te chains are not equidistant. These observations from STM show that the $M_x$ mirror symmetry is broken, which results in non-centrosymmetry. $\mathbf{q}_{CDW}$ has been estimated from the satellite spots observed in the Fourier transform of the STM image as well as from the low energy electron diffraction (LEED) pattern that also shows CDW related satellite spots (encircled in Fig. 1c) besides the ($1 \times 1$) spots; see Supplementary Note 1 and Supplementary Fig. 4. The non-centrosymmetry in the CDW state of LaTe$_3$ is also demonstrated by the peaks of $B_1$ symmetry mode[34] in the Raman spectrum in Supplementary Fig. 5.

The high symmetry directions are shown in the BZ of the 7f structure in Fig. 1d inscribed within the non-CDW BZ. Since the BZ is related to the ordering of the lattice constants of the conventional cell, its comparison with the primitive unit cell is shown in Supplementary Fig. 6a, b. In our notation, the horizontal plane in the BZ is represented by $k_x$-$k_z$ and $\mathbf{q}_{CDW}$ is oriented along $k_z$. The CDW BZ, containing features such as the primitive reciprocal lattice vectors and all the pertinent high symmetry points and directions as well as their coordinates, is depicted in Supplementary Fig. 6c. In what follows, we present the ARPES data in the next subsection that is subsequently compared with the effective band structure based on the 7f structure. The DFT bands for the 7f structure without (w/o) SOC are discussed next, followed by the band structure with SOC. The existence of the KNL is established and ARPES bands along the high symmetry directions are discussed in the last two subsections. Finally, in the Discussion section, we confirm the existence of the KNL and the spinless nodal line for the 29f structure.

### Crossing of the bilayer-split shadow and main bands from ARPES

An $E(k_z)$ ARPES intensity plot in a generic direction parallel to $\Gamma Z$ at $k_x = 0.68$ Å$^{-1}$ i.e., near the $X$ point (the length of $\Gamma X$ being 0.737 Å$^{-1}$) is shown in Fig. 2a in the CDW state (see Methods for the experimental details). The direction of the ARPES measurement is shown by the red line denoted by a in Fig. 2g. Figure 2a shows two main bands (inner and outer) centered around $X$ that cross $E_F$ at $k_z = \pm 0.15$ and $\pm 0.21$ Å$^{-1}$, respectively. The outer main band disperses down to binding energy ($E$) of about 1.25 eV, while the inner band has a nearly flat bottom at ∼0.8 eV. It is interesting to note that a relatively weaker band centered around $k_z = \pm 0.41$ Å$^{-1}$ is a replica of the main band shifted by $\mathbf{q}_{CDW}$ (= 0.28$\mathbf{c}^*$ = 0.41 Å$^{-1}$), as shown by two horizontal white dashed arrows. This replica band crosses $E_F$ at $k_z = 0.21$ and 0.61 Å$^{-1}$ and disperses down to $E$ ∼1.25 eV. No shift in its position along $E$ compared to the main band is observed, which indicates that it is related to the initial state CDW superlattice[45]. It has been referred to in the literature as the shadow band[36,45,46]. The signature of the shadow band is also evident in the Fermi surface shown in Fig. 1e, where shadow Fermi surface branches appear at a separation of $\mathbf{q}_{CDW}$ from the main branches, as shown by black dashed arrows in the metallic region around the $X$ point parallel to the $\Gamma Z$ direction. In the Supplementary Note 2, a

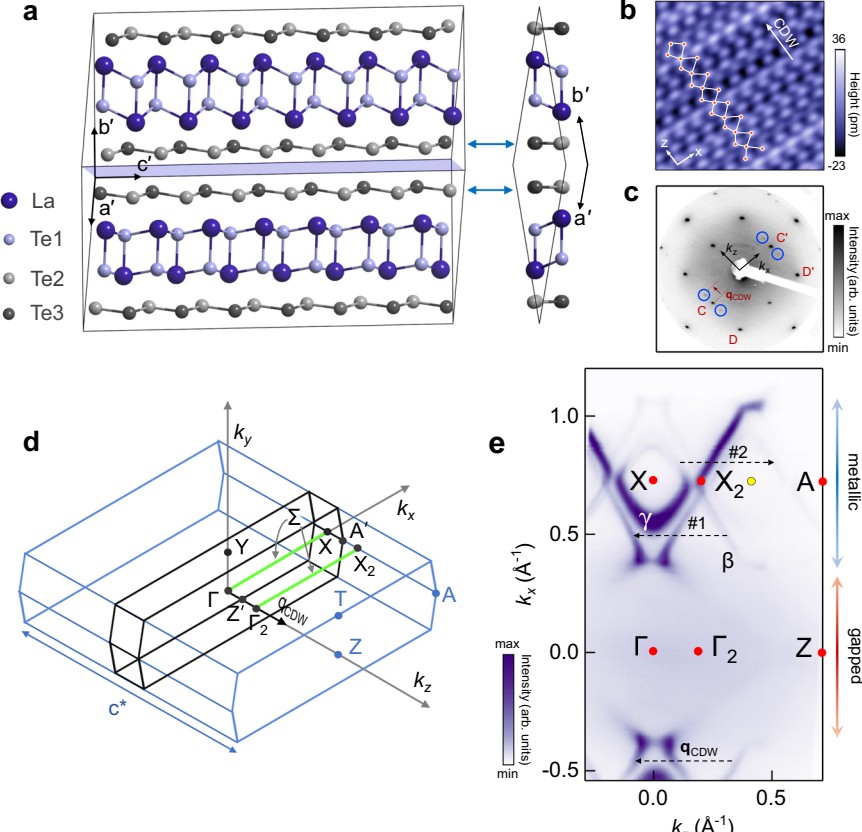

**Fig. 1 | Structure, Brillouin zone and the Fermi surface of LaTe₃ in the CDW state. a** The 7-fold (1 × 1 × 7) modulated primitive unit cell of LaTe₃ comprising of 56 atoms with wave vector $\mathbf{q}_{CDW} = \frac{2}{7}\mathbf{c}^*$ viewed perpendicular (left) and parallel (right) to the $\mathbf{c}'$ direction. The lattice constants are $a' = b' = 13.256$ Å, $c' = 30.778$ Å with $\alpha' = \beta' = 90°$, and $\gamma' = 160.99°$. The cleavage plane (light blue) occurs in between the two weakly interacting Te layers indicated by the blue double-sided arrows. **b** High resolution scanning tunneling microscopy topography image (4.2 nm × 4.2 nm) obtained with bias voltage of 0.2 V and a tunneling current of 0.4 nA. The overlaid orange circles represent the Te atom positions. The white dashed lines show the average positions of the adjacent Te chains. **c** Low energy electron diffraction pattern, measured with 77 eV primary beam energy, shows the reciprocal space in the $k_x$-$k_z$ plane (black arrows). C, D, C′, and D′ represent the main spots; the CDW satellite spots are highlighted by blue circles. **d** The CDW Brillouin zone (BZ) (black) is shown within that of the non-CDW state (light blue). The high symmetry points[83] are indicated in these respective colors. Γ, Y and X points coincide for both. The Σ lines represented by ΓX and Γ₂X₂ in the 2ⁿᵈ BZ are shown in green color. See Supplementary Fig. 6c for the other high symmetry points and directions in the CDW BZ. **e** The Fermi surface in the CDW state measured by ARPES, where the length of the black dashed arrows (#1, #2) that join the shadow branches with the main branches of the Fermi surface represents $\mathbf{q}_{CDW}$. The colorbar shows the intensity in arbitrary units (arb. units).

discussion about the Fermi surface and $\mathbf{q}_{CDW}$ obtained from the average separation of the shadow and main branches is provided. $\mathbf{q}_{CDW} = 0.28 \pm 0.005\mathbf{c}^*$ determined in this way agrees with the values obtained from STM and LEED (see Supplementary Note 1), as well as that from transmission electron microscopy ($\mathbf{q}_{CDW} = 0.28 \pm 0.01\mathbf{c}^*$)[47]. However, the accuracy of these approaches for calculating $\mathbf{q}_{CDW}$ is substantially worse compared to x-ray crystallography[33].

In Fig. 2a, the shadow and the main bands resemble an inverted V and meet each other close to $E_F$ at $k_z = \pm 0.21$ Å⁻¹ (highlighted by a green dashed oval). This region is shown in an expanded scale as a curvature plot in Fig. 2e, where the red and black dashed lines suggest a possible crossing of nearly linear bands at $E_F$. The Fermi velocities of these bands, determined using the expression ($\frac{1}{\hbar}\frac{dE}{dk_z}$), turn out to be $(1.2 \pm 0.05) \times 10^6$ and $(1 \pm 0.05) \times 10^6$ m/s for the shadow and main bands, respectively. These values are comparable to graphene ($1 \times 10^6$ m/s[48]), indicating large mobility of LaTe₃ in agreement with recent report from Hall conductivity measurements[30].

To further probe these bands and their possible crossing, ARPES was performed over a range of $k_x$ out of which three representative plots (along the red lines parallel to a up to d in Fig. 2g) are shown in Fig. 2b–d. Interestingly, as $k_x$ decreases, both the main and the shadow bands spread out in $k_z$ and the crossings—highlighted by green dashed ovals—shift below $E_F$ to larger $E$. However, note that their $k_z$ position

remains unchanged around $X_2 \approx \pm\frac{1}{2}\mathbf{q}_{CDW}$. In Fig. 2b–d, splitting in both the shadow and the mains bands are observed. This is also observed in Fig. 2e, as shown by the double-sided red arrows. This is dubbed as bilayer splitting and has been reported in other RTe₃ members[36,49,50], bilayer graphene[51], and cuprate superconductors[52]. In Supplementary Note 3, from the decrease in the bilayer splitting calculated by DFT as the separation of the Te bilayers increases, we confirm that it is related to the coupling between the two adjacent Te layers. Moreover, it is observed for both non-CDW and CDW states of LaTe₃ and does not appear in LaTe₂ that has a single layer of Te but exhibits CDW[53,54]. These results indicate that the bilayer splitting would occur in a Te bilayer independent of the CDW.

Both the bilayer-split shadow bands—denoted by *su* (*sd*) for smaller (larger) *E*, see Fig. 2f—are shifted by $\mathbf{q}_{CDW}$ from the corresponding main bands denoted by *mu* and *md*. This is shown by the white dashed horizontal arrows in Fig. 2a, b. This leads to formation of four crossings between the main and the shadow bands, as shown in Fig. 2f. A stack of $k_z$-$k_x$ isosurface plots with *E* varying from 0 to 0.6 eV with the crossings visible at *E* = 0.5 eV (a zoomed curvature image shown) and a stack of momentum distribution curves (MDCs) taken near the crossings in Fig. 2c are shown in Supplementary Fig. 13a, b.

The crossings are denoted by left (*L*, highlighted by blue rectangle, crossing of *mu* and *sd* i.e., *mu* ⊗ *sd*), right (*R*, orange rectangle,

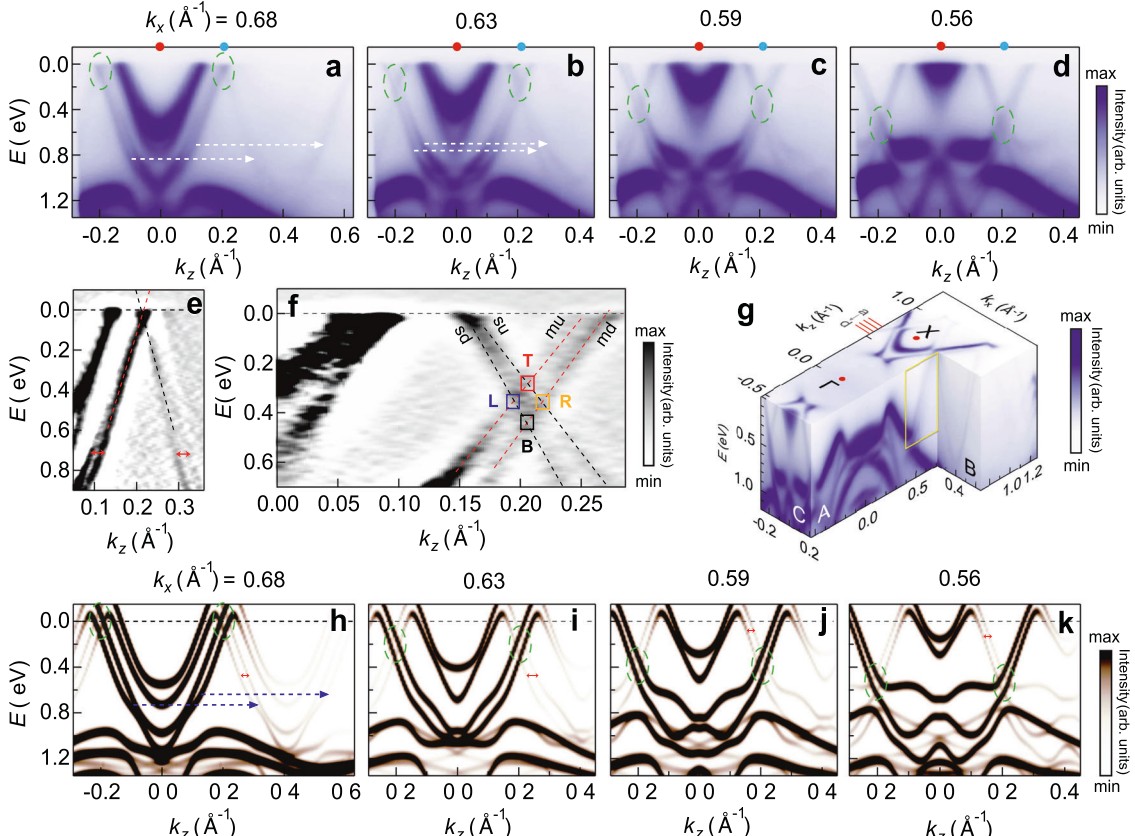

**Fig. 2 | Crossings between the bilayer-split shadow and the main bands.** $E(k_z)$ ARPES intensity plots measured at 100 K for $k_x$ = **a** 0.68, **b** 0.63, **c** 0.59, and **d** 0.56 Å$^{-1}$. The red and blue dots at the top of the panels indicate the $k_z$ positions of the $\Gamma X$ and $\Gamma_2 X_2$ high symmetry lines. The white dashed arrows in panels **a** and **b** represent the $\mathbf{q}_{CDW}$ and the band crossing regions are highlighted by green dashed ovals. $E = 0$ represents the $E_F$. These panels share same $y$ axis and colorbar, the latter is shown on the right of panel **d**. **e** A zoomed view of the near $E_F$ region of panel **a** around $k_z$ = 0.21 Å$^{-1}$, the dashed red (black) line that represents the main (shadow) band is obtained by curve fitting (see Methods). The double-sided red arrows show the bilayer splitting. **f** Curvature plot of a part of panel **c** shown in an expanded scale: the $L$, $B$, $R$ and $T$ crossings are highlighted by blue, black, yellow and red rectangles, respectively. The $T$ and $B$ crossings occur on the $\Gamma_2 X_2$ line. The dashed red [black]

lines representing the bilayer split main ($mu$,$md$) [shadow ($su$,$sd$)] bands are obtained by curve fitting as in **e**. **e**, **f** share the same colorbar. **g** Cut $A$ at $k_z$ = 0.216 of an $E$-$k_z$-$k_x$ ARPES intensity plot shows the dispersion of the $R$ crossing and the $mu$ and $sd$ bands within the yellow rectangle. A shadow band is observed in the cut $B$. The ARPES cuts measured along the red lines a-d on the $k_x$ axis are shown in panels **a**–**d**. **h**–**k** The effective band structure (EBS) along $k_z$ obtained by band unfolding at similar $k_x$ values as in **a**–**d**, respectively. The intensity scale represents the spectral weights calculated from band unfolding and the size of the dots is related to the instrumental resolution. **h**–**k** share same $y$ axis and colorbar, the latter is shown on the right of **k**. The double-sided red arrows indicate the bilayer splitting and the blue arrows in panel **h** represent the $\mathbf{q}_{CDW}$. The band crossing regions are highlighted by green dashed ovals.

$md \otimes su$), top ($T$, red rectangle, $mu \otimes su$), and bottom ($B$, black rectangle, $md \otimes sd$) (Fig. 2f). While Fig. 2a–d shows the band dispersion at discrete $k_x$ values, we show a continuous dispersion of the $R$ crossing in cut $A$ of Fig. 2g. Here, the $E(k_x)$ at $k_z = 0.216$ Å$^{-1}$ shows the loci of the $R$ crossing within the yellow rectangle.

ARPES with different photon energies shows negligible $k_y$ dependence of the crossings. For example at $k_x = 0.58$ Å$^{-1}$, Supplementary Fig. 14a–d shows that their position (highlighted by green dashed ovals) remains almost unchanged with $k_y$. This is summarized in Supplementary Fig. 14e through a $k_y$-$k_z$ map at $E = 0.39$ eV, where the two crossings at $k_z = \pm 0.2$ Å$^{-1}$ (marked by yellow arrows) show almost no change with $k_y$, indicating the quasi-2D nature of LaTe$_3$.

### Effective band structure from DFT compared to ARPES
Multiple crossings indicated by ARPES is a surprising result since hybridization gap is generally expected at the band touching points between the Bloch states connected by $\mathbf{q}_{CDW}$[50,55]. The $E(k_z)$ bands calculated for the 7f structure at different $k_x$ as in the experiment are shown in an extended zone scheme spread over the non-CDW BZ in Supplementary Fig. 15. Many bands are observed in the CDW state due to band folding. Their complexity hides the influence of the CDW on the electronic bands and impedes their interpretation and direct

comparison with ARPES. So, an effective band structure (EBS) has been calculated by unfolding the bands in the non-CDW BZ[56–58]. In Fig. 2h–k, EBS shows the distribution of states as a function of their energy and momenta with appropriate spectral weight, where broadening similar to the experiment has been applied. Variations in the spectral weight is evident resulting in dissimilar EBS in the different CDW BZs.

The importance of the EBS calculation is that despite the CDW effect being small with the modulation amplitude being only ~ 4.5% of the average Te2-Te3 distances, the EBS reveals the shadow bands. These are shifted from the main bands by $\mathbf{q}_{CDW}$ (blue dashed arrows) in excellent agreement with ARPES (compare Fig. 2h–k with Fig. 2a–d). In contrast, the bands in the non-CDW state, where obviously the CDW amplitude is zero, shows only the main bands (Supplementary Fig. 16a). An analysis of the orbital character establishes that the shadow bands (as well as the main bands) are of predominantly in-plane Te $p_x$-$p_z$ character, see the Supplementary Fig. 17a–f. The out-of-plane $p_y$ character becomes slightly significant only for $E > 0.6$ eV, and the contributions from the La-Te1 block is negligible. However, transfer of electrons from La to the Te net determines the band filling in the Te2-Te3 layer and this has been calculated using the Bader charge analysis (see Supplementary Note 4 including Supplementary Table 1 and refs. [59–61]).

From Fig. 2h, we find that the outer branch of the shadow band disperses towards $E_F$ and crosses the main band around $k_z = \pm\,0.21\,\text{Å}^{-1}$ (highlighted by a dashed green oval). The inner branch of the shadow band which disperses to about 0.6 eV, although more prominent in the EBS, has its counterpart in experiment in the curvature plot shown in Supplementary Fig. 16b. A bunch of bands that disperse weakly between 1.15 to 1.6 eV is observed in both theory and experiment at similar $E$. In Fig. 2i–k, for progressively smaller $k_x$ values, the EBSs portray the crossings ($L$, $R$, $T$ and $B$) at similar $E$-$k_z$ as observed in ARPES (compare Fig. 2i–k with Fig. 2b–d). The observation of four distinct crossings is related to the bilayer splitting that is also observed in the EBS (horizontal red double arrows in Fig. 2h–k). Also note that the bilayer splitting ($\Delta k_z$) increases with $E$ for the different $k_x$ and this trend is observed in both experiment and theory (see Supplementary Table 2). Similar variation of $\Delta k_z$ has been recently reported for NdTe$_3$[62]. The agreement of ARPES and EBS is also good at larger $E$: the main bands around 0.8 eV become flatter and move to lower $E$. Important to note is that from ARPES the crossings appear to be gapless (Fig. 2a–f) within the experimental and lifetime broadening. Although this is supported by the EBS (Fig. 2h–k), note that the calculations have been performed here with a $k$ step size ($\delta k_z$) of $6 \times 10^{-3}\,\text{Å}^{-1}$, and the broadening is comparable to ARPES, both of which might conceal potential presence of minigaps. In the subsequent subsections, we show the DFT bands calculated with smaller $\delta k_z$ to further probe the nature of the crossings.

## Spinless nodal lines formed by the $L$ and $R$ crossings

The $E(k_z)$ bands from DFT with small $\delta k_z$ ($= 5 \times 10^{-4}\,\text{Å}^{-1}$) w/o SOC show that the $L$ and $R$ crossings are gapless (Fig. 3a). This is reconfirmed in Fig. 3b, c by the bands calculated with even smaller $\delta k_z$ ($= 1 \times 10^{-5}\,\text{Å}^{-1}$). It

is interesting to note that these crossings occur at generic points of the BZ in the $k_x$-$k_z$ plane and the bands involved are linear (Fig. 3a–c). The unfolded EBS in Supplementary Fig. 18 shows nice agreement with the ARPES intensity plots in Fig. 2c, f and demonstrate the linearity of the bands around $L$ and $R$. The velocities calculated from their slopes are similar to that obtained at $E_F$ from Fig. 2e. Although these bands originate from the in-plane $p$ orbitals with small difference in contributions from the $p_x$ and $p_z$ orbitals as shown in Supplementary Fig. 17a–f, $md$ and $sd$ bands belong to $M_1$ irreducible band representation (irrep), while $mu$ and $su$ belong to $M_2$ irrep. Thus both $L$ and $R$ crossings are formed by bands belonging to different irreps (Fig. 3a–c). Note that band folding results in two more relatively flat bands in Fig. 3a, i (also shown in Supplementary Fig. 18a, c where these are indicated by red arrows). These bands are not detected in the unfolded EBS because of their reduced spectral weight (Supplementary Fig. 18b, d) and are also not observed in ARPES (Fig. 2c, f).

The crossings disperse with $k_x$ and at larger $k_x = 0.685\,\text{Å}^{-1}$ compared to $0.59\,\text{Å}^{-1}$, $L$ and $R$ crossings traverse the $E_F$ (Supplementary Fig. 19a–c). In fact, calculations for a series of $k_x$ values establish that the gapless linear crossings occur over an extended range of the $E$-$k$ space (Supplementary Fig. 20). The loci of each crossing form a continuous curve in the momentum space that has been referred to as a nodal line. A direct comparison of the ($E$, $k_x$) cuts from ARPES (Fig. 3f, g for $R$ and $L$, respectively) shows that the positions of both the nodal lines are in excellent agreement with DFT. Both disperse between ($E$, $k_x$) = (0 eV, ~ 0.7 Å$^{-1}$) to (~ 0.6 eV, ~ 0.5 Å$^{-1}$) with $k_z$ at 0.195 and 0.22 Å$^{-1}$ for $L$ and $R$, respectively. Thus, the crossings appear within an energy window of $E$ ~ 0.6 eV to the $E_F$. Their projections in the $k_x$-$k_z$ plane form a pair of approximately parallel nodal lines that are 0.2 Å$^{-1}$ in length and appear at a separation of $k_z$ ~ 0.02 Å$^{-1}$ in a general direction on this

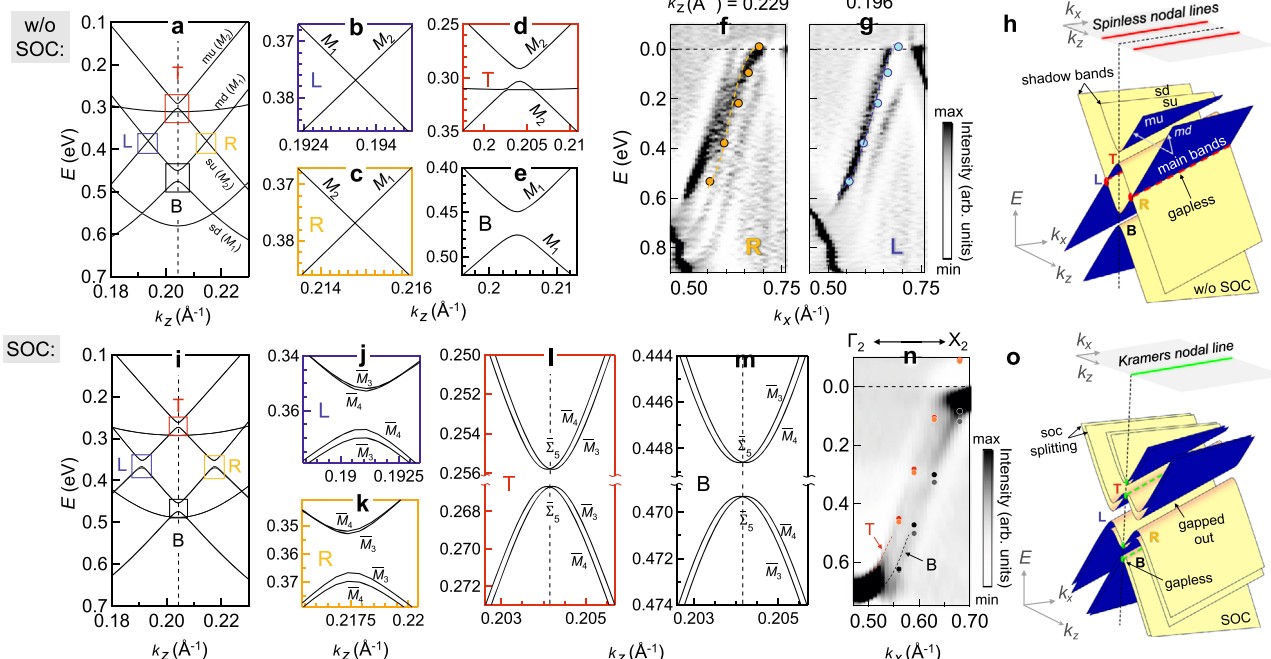

**Fig. 3 | Band crossings and the Kramers nodal line (KNL).** a $E(k_z)$ bands at $k_x =$ 0.59 Å$^{-1}$ without (w/o) spin-orbit coupling (SOC) for the 7f structure. The irreducible representations are shown. The vertical dashed line represents the $k_z$ point on $\Gamma_2 X_2$ i.e., the $\Sigma$ line in the 2$^{nd}$ BZ (see Fig. 1d). Zoomed colored rectangles of **a** show the bands around **b** $L$, **c** $R$, **d** $T$, and **e** $B$. Comparison of ARPES and DFT for **f** $R$ at $k_z =$ 0.229 Å$^{-1}$ and **g** $L$ at $k_z =$ 0.196 Å$^{-1}$. The positions of these crossings obtained from DFT (orange and blue filled circles for $R$ and $L$, respectively) are superimposed. The dashed orange and blue curves serve as guide to the eye. **h** A schematic representation of the gapless $L$ and $R$ crossings in the $E$-$k_z$-$k_x$ space (red

dashed lines) and their projection on the $k_x$-$k_z$ plane showing the spinless nodal lines (thick red lines on both sides of the $\Sigma$ line). **i–m** Same as **a–e** except that the calculations are performed with SOC. **n** $E(k_x)$ ARPES intensity plot at $k_z =$ 0.204 Å$^{-1}$ (dashed red and black curves serve as guide to the eye) compared with the positions of the crossings from DFT for $T$ (red, light red circles) and $B$ (black, gray circles). **o** A schematic representation of the four crossings (green dashed lines) related to the upper and lower branches of $T$ and $B$. The KNL appears along the $\Sigma$ line and is denoted by a green thick line on the $k_x$-$k_z$ plane.

plane (red solid lines in Fig. 3h). These appear parallel to but on either sides of $\Gamma_2 X_2$ that occurs at $k_z = 0.204$ Å$^{-1}$. These nodal lines are formed by two fold crossings of nondegenerate bands in the absence of spin and so are referred to as spinless nodal lines. The existence of these nodal lines is shown for the 29f structure of LaTe$_3$ in the Discussion section.

With inclusion of SOC, it is intriguing to find that both the $L$ and $R$ crossings (and hence the nodal lines) are entirely gapped out by minigaps of 14 meV (for $k_x = 0.59$ Å$^{-1}$) to 17 meV (for $k_x = 0.685$ Å$^{-1}$) into upper and lower branches, as shown in Fig. 3i–k and Supplementary Fig. 19g, h. A schematic is shown in Fig. 3o. The gap is formed by hybridization of bands belonging to same one dimensional double group irrep involving both $\overline{M_3}$ and $\overline{M_4}$.

Note that non-centrosymmetry should lift the SU(2) spin degeneracy. This is visible through the spin splitting of both the upper and lower branches in Fig. 3j, k. These spin-split bands belong to different irreps $\overline{M_3}$ and $\overline{M_4}$ and do not cross each other. The splittings for both $L$ and $R$ are $k$-dependent, it is maximum at the extrema (~3 meV for lower and 1 meV for the upper branch) and decreases away from it.

### Evidence of Kramers nodal line from the $T$ and $B$ crossings

In the case of $T$ and $B$ crossings, the shadow and main bands belong to the same irrep, leading to hybridization-related minigaps w/o SOC and the creation of upper and lower branches (Fig. 3a, d, e). The inclusion of SOC provides a fascinating outcome: the spin-split bands in both the upper and lower branches of $T$ and $B$ exhibit gapless crossings (Fig. 3l, m), which contrasts the bands at $L$ and $R$ (Fig. 3j, k). Each branch of $T$ and $B$ crosses at the same $k_z = 0.204$ Å$^{-1}$ (dashed lines in Fig. 3i, l, m) resulting in four crossings. This value of $k_z$ is special, since it falls on the $\Gamma_2 X_2$ line of the CDW BZ (i.e., $\Gamma X$ or the $\Sigma$ line) (Fig. 1d). At $k_x = 0.59$ Å$^{-1}$, the four crossings appear at different $E$: 0.256 (0.267) eV for the upper (lower) branch of $T$, and 0.448 (0.469) eV for upper (lower) branches of $B$.

These gapless crossings are observed over a range of $k_x$, as in the case of $L$ and $R$. For example, at $k_x = 0.685$ Å$^{-1}$, the crossings disperse to smaller $E$ and in this case both the branches of $T$ ($B$) are above (below) the $E_F$ (Supplementary Fig. 19f, i, j). Significantly, the crossings always appear at same $k_z$, i.e., along the $\Sigma$ line, suggesting that the crossings may be enforced by the lattice symmetries along this direction.

The band irreps with SOC shown in Fig. 3l, m are as follows: the crossing belongs to a double valued irrep that is two dimensional ($\overline{\Sigma}_5$), while away from it, the spin-split bands have one dimensional ($\overline{M}_3$ or $\overline{M}_4$) representation. Significantly, the $\Sigma$ line that emerges from the $\Gamma$ point has the little group that is isomorphic to $C_{2v}$ point group and has symmetries such as the two fold rotation about the $k_x$- axis denoted by $C_{2x}$: $\{2_{100}|0, 0, 0\}$, glide reflection perpendicular to the $y$ axis in the $k_x$-$k_z$ plane followed by a translation of $\frac{1}{2} c$ given by $\tilde{M}_y$ : $\{m_{010}|0,0,\frac{1}{2}\}$ and an off-centered mirror perpendicular to the $k_z$ axis $M_z$ : $\{m_{001}|0,0,\frac{1}{2}\}$. $\Gamma$ is a TRIM point, where according to the Kramers theorem, each band is at least doubly degenerate. The little group of $\Sigma$ is related to that of $\Gamma$ by the compatibility relations. We find that $\Gamma$ and $\Sigma$ are represented by two dimensional double-valued irreps: $\overline{\Gamma}_5$ and $\overline{\Sigma}_5$, respectively. These representations are similar, as shown in Supplementary Table 3[63]. The $\Sigma$ line passes through the $X$ point with coordinates $(0.257, 0.257, 0)$ and meets the TRIM point $Y_2$ $(0.5, 0.5, 0)$ in the next BZ (see Supplementary Fig. 21). $Y_2$ also belongs to double-valued two dimensional irrep $\overline{Y_{25}}$ that is same as $\Gamma$ and $\Sigma$ (Supplementary Table 3). The condition that at the TRIM points the representations should be time reversal invariant is satisfied since both $\overline{\Gamma}_5$ and $\overline{Y_{25}}$ are pseudo-real[63]. Thus, $\overline{\Gamma}_5$-$\overline{\Sigma}_5$-$\overline{Y_{25}}$ is able to support two fold degeneracy of the bands along $\Gamma X Y_2$ i.e., the $\Sigma$ line. This is a Kramers nodal line (KNL) that occurs along the $\Sigma$ line in the mirror-invariant $k$-plane in the presence of TRS, and the additional rotational symmetry constrains the KNL along a high symmetry direction[25]. Xie et al.[25] have proved the existence of KNL along the $C_2$ rotational axis ($\{2_{100}|0, 0, 0\}$), which in our case is along $k_x$ that lies in

the $k_x$-$k_y$ mirror plane ($\{m_{001}|0,0,\frac{1}{2}\}$). In Fig. 3o, the projection of the loci of the crossings of the four pairs of bands in the $E$-$k_z$-$k_x$ space on the $k_x$-$k_z$ plane shows the KNL (thick green line) that enforces the crossings.

The bands in a plane that cuts the KNL perpendicularly−as is the case in Fig. 3i−have been described in the literature as two-dimensional massless Dirac Hamiltonian with the Berry curvature concentrated at the crossing[25]. These authors demonstrated that the Berry phase around a KNL is quantized as $m\pi$ mod $2\pi$. In case of quadratic and cubic dispersion of these bands, the crossing has been dubbed as a higher-order Dirac point[25]. In the present case, both the upper and lower branches of $T$ and $B$ exhibit quadratic dispersion close to the crossings, as is evident in Fig. 3l, m. Consequently, the gapless crossings in LaTe$_3$ that are associated with the KNL are higher-order Dirac points. The gapless crossings and their quadratic dispersion are demonstrated for the 29f structure of LaTe$_3$ in the Discussion section.

### ARPES and DFT along the KNL and other directions

In Fig. 4a, the bands calculated along the KNL (i.e., $\Gamma X$) show two pairs of degenerate bands related to the $T$ and $B$ crossings, highlighted by green shading, and zoomed in the insets $i, ii$. The ARPES intensity plot in Fig. 3n represents these bands. These are also identified by the degenerate crossings of bands in the $E(k_z)$ direction enforced by the KNL (Fig. 3l, m). The existence of the KNL between 0.5 and 0.7 Å$^{-1}$ along $k_x$ is affirmed by ARPES through an excellent agreement of the positions of the crossings obtained from DFT (filled circles) that are overlaid on the experimental data in Fig. 3n. The agreement is also evident in the Supplementary Fig. 22, where Fig. 3i is superposed on Fig. 2f. However, the energy separations between the upper and lower branches of both $T$ and $B$ are too small to be resolved by ARPES. Similarly, the quadratic nature visible in a tiny $k_z$ range shown in Fig. 3l, m is not distinguished, see the regions enclosed by the red and black rectangles in Supplementary Fig. 22. Nonetheless, ARPES is consistent with DFT, and both show that the crossings disperse from $E$ ~ 0.6 eV at $k_x$ ~ 0.55 Å$^{-1}$ and traverse the $E_F$ at $k_x = 0.65$–0.7 Å$^{-1}$.

We find that every band along $\Gamma X$ is degenerate due to the double degeneracy enforced by the KNL (see for example the insets $i$-$v$ of Fig. 4a) and belong to two dimensional $\overline{\Sigma}_5$ irrep. Besides the $T$ and $B$ related bands discussed above, there are other bands that cross $E_F$ in Fig. 4a. These bands−numbered as $1, 2,$ and $3$ and highlighted by orange shading−have not been observed in the ARPES along $\Gamma_2 X_2$ ($k_z = 0.204$ Å$^{-1}$) due to their low spectral weight at this $k_z$. However, these three bands are clearly visible in the ARPES intensity plot along $\Gamma X$ ($k_z = 0$ Å$^{-1}$) in Fig. 4b and disperse across the $E_F$ in splendid agreement with the EBS along the same direction in Fig. 4c. At larger $E$ other bands (numbered as $4$-$12$) in the EBS along $\Gamma X$ are in very good agreement with ARPES (Fig. 4b). On the other hand, the bands around the $T$ and $B$ crossings both along and perpendicular to the KNL have negligible spectral weight around $\Gamma X$ and so are observed neither in EBS nor ARPES in Fig. 4b, c. However, the band structure shows the crossings formed by band folding in every BZ, see Supplementary Fig. 15, where the crossing region is highlighted by red ovals. This shows the importance of performing ARPES over multiple CDW BZs in the direction of $\mathbf{q}_{CDW}$ to decipher the influence of the CDW on the electronic band structure.

Perpendicular to the KNL along $\Gamma Z$ ($\Lambda$), the bands are represented by one dimensional irreps (Supplementary Fig. 21 and Supplementary Table 4). Since the little group along this direction has lesser symmetry than $C_{2v}$, degeneracy is not enforced. This is shown by the spin-splitting along $\Gamma Z$ in the zoomed insets $iv, v$ of Fig. 4a. This is also true for the bands calculated along various other high symmetry directions (Supplementary Fig. 23), where the insets show degeneracy along $\Gamma X$ and splitting along the other directions. In the $\Gamma Z$ direction no bands are found to cross the $E_F$, and this is corroborated by ARPES as well as EBS. A hybridization related gap is observed with bands $13$ and $14$ being

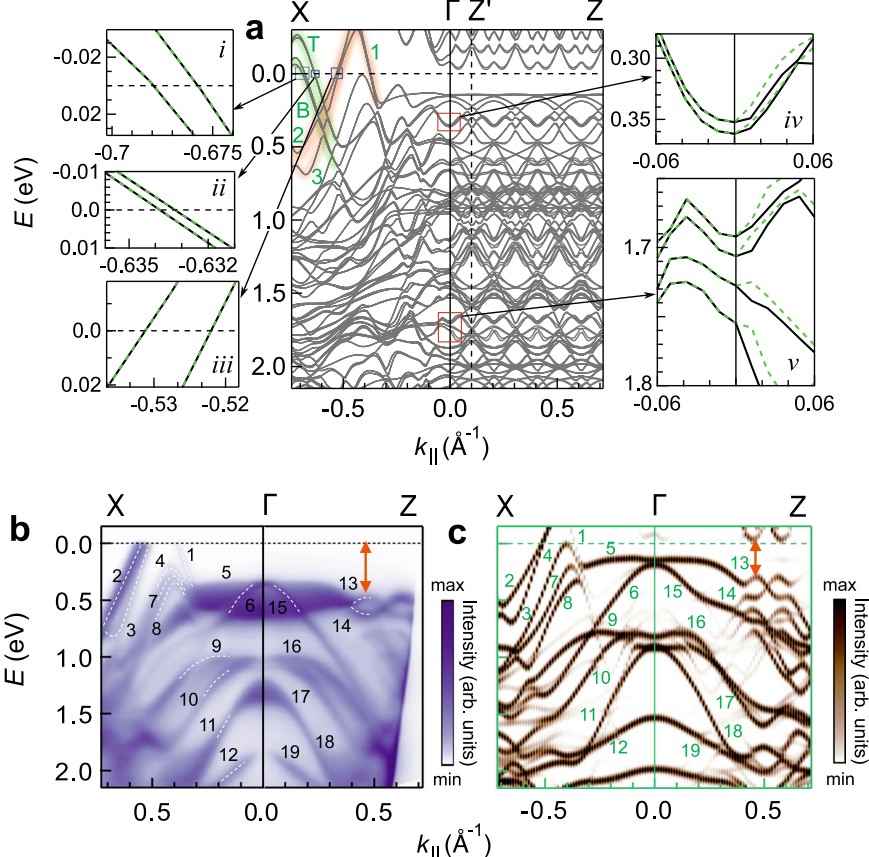

**Fig. 4 | DFT and ARPES along and perpendicular to the KNL. a** The band structure of LaTe$_3$ in the CDW state with the 7f structure along $X\Gamma Z'Z$. On the left, zoomed regions from the blue rectangles show the two fold degeneracy (dashed-green and black) of the bands along the KNL. On the right, zoomed regions from the red rectangles show the spin-splitting along the $\Gamma Z$ direction in contrast to $\Gamma X$. The bands that cross $E_F$ are highlighted by orange and green shading. **b** ARPES intensity plot measured using photon energy of 24.4 eV, and **c** the EBS obtained from DFT towards $\Gamma X$ and $\Gamma Z$. The white dashed curves in **b** serve as guide to the eye for some of the bands that are numbered from 1 to 19. The orange double sided arrows indicate the CDW gap. All the panels share the same $x$ axis.

the highest occupied ones (orange double arrow in Fig. 4b, c). This gap has been referred to as the CDW gap in LaTe$_3$[36] and other RTe$_3$ systems[45,49,50,62,64]. A discussion about the variation of the CDW gap of LaTe$_3$ with $k_x$ and $k_y$ is provided in the Supplementary Note 5.

## Discussion

As discussed in the previous section, DFT calculations for the 7f structure have demonstrated the existence of the nodal lines. In addition, an outstanding overall agreement with ARPES is obtained, which can be related to the small difference of the atom positions compared to x-ray crystallography (see Supplementary Fig. 2b). Nevertheless, it is crucial to establish the nodal lines for the 29f structure of LaTe$_3$, whose atom positions are indistinguishable from x-ray crystallography[33] within the experimental accuracy, as shown in Supplementary Fig. 2a.

Here, the results of our DFT calculations with SOC for the 29f structure at $k_x = 0.59$ Å$^{-1}$ in Fig. 5a, b show that the KNL related crossings at $T$ and $B$ are intact, and their dispersion around the crossings is quadratic, as in the 7f structure. The crossings appear at 0.197 Å$^{-1}$ that is slightly different from 7f (0.204 Å$^{-1}$) because the size of their BZs are not integral multiple of each other (BZ of 29f is $\frac{7}{29}$ times reduced along $k_z$). To show the nodal line character of the KNL, we show these crossings at another $k_x$ value of 0.685 Å$^{-1}$ in Supplementary Fig. 24a, b.

The KNL indicated by these crossings appears along the $\Sigma$ line. The spin degeneracy of the bands along $\Gamma X$ confirms this, whereas, in contrast, the degeneracy is lifted along $\Gamma Z$. This is shown in Fig. 5e, f which are zoomed regions enclosed by red rectangles in

Supplementary Fig. 25, where zoomed regions from the blue rectangles show this for other $E$ ranges. Thus, the existence of the KNL is demonstrated for the CDW state of LaTe$_3$ with the 29f structure. Its behavior is similar to that of the 7f structure since both possess the same symmetry.

DFT calculations w/o SOC for the 29f structure show the existence of the spinless nodal lines related to the $L$ and $R$ crossings (see Fig. 5g, h for $k_x = 0.59$ Å$^{-1}$ and Supplementary Fig. 24c, d for $k_x = 0.685$Å$^{-1}$), as in the 7f structure. These spinless nodal lines comprise of crossings of nondegenerate bands with distinct irreps that are linear and traverse the $E_F$ with high Fermi velocity. These are however gapped out with SOC (Fig. 5c, d). This behavior resembles non-centrosymmetric topological nodal line semimetals such as pinictides e.g., CaAgAs, where the SOC-induced gap results in a topological insulator phase[65,66]. Whether the SOC-induced gap in LaTe$_3$ has a topological character is an open question that would require further research.

The appearance of two distinct types of nodal lines (spinless and KNL) discussed above can be attributed to the bilayer splitting, that splits both the shadow and the main bands belonging to different irreps. The crossings of the CDW-induced shadow with the main bands are enforced by the KNL. The crossings occur from $E \sim 0.6$ eV and disperse in $E$ to traverse the $E_F$ as $k_x$ increases (Fig. 3n). This dispersion in $E$ is related to the dispersion of the main band (the crossing of the main band with $E_F$ moves to larger $k_z$ as $k_x$ decreases, see Fig. 2a–d, h–k) coupled with the constraint that the shadow band is separated from it by $q_{CDW}$. Other spin degenerate bands also cross the $E_F$ along the KNL. So, we characterize LaTe$_3$ to be a KNL metal in

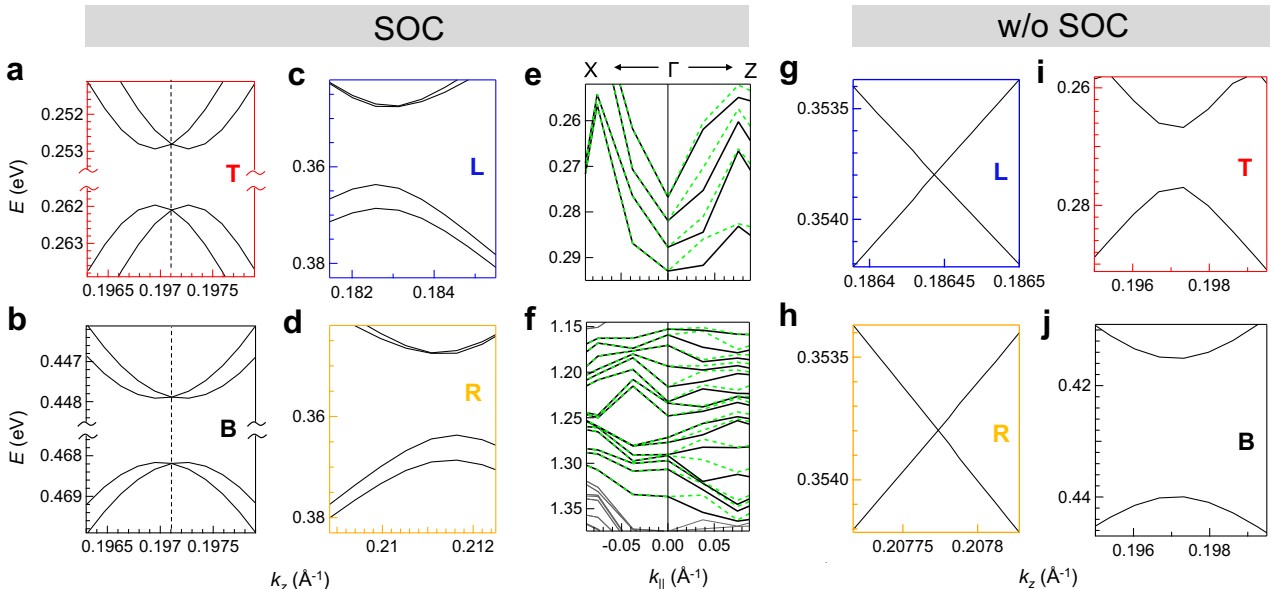

**Fig. 5 | Band crossings for 29 fold LaTe₃ at $k_x$ = 0.59 Å⁻¹.** Band dispersion with SOC around **a** $T$, **b** $B$, **c** $L$, and **d** $R$. **e, f** Two fold degenerate bands (dashed-green and black) along $\Gamma X$ and splitting along $\Gamma Z$ shown in magnified scale. Band dispersion w/o SOC around **g** $L$, **h** $R$, **i** $T$, and **j** $B$.

the CDW state based on multiple bands crossing the $E_F$ both along and perpendicular to the nodal line.

In conclusion, the identification of the shadow band from both ARPES and DFT is the foundation of our study. Its interaction with main band—studied using DFT for an incommensurate system with a realistic structure within experimental accuracy—resulted in the revelation of CDW-induced KNL in LaTe₃, a non-centrosymmetric, quasi-2D, TRS-preserving incommensurate material. This discovery is the cornerstone of our investigation, which we believe will inspire further research in the fascinating field of CDW materials.

## Methods

### Experimental

Single crystals of LaTe₃ with residual resistivity ratio [$\rho$(300 K)/$\rho$(2 K)] of ~270[30] were grown by tellurium flux technique, where high-purity La and Te were mixed in a molar ratio of La₀.₀₂₅Te₀.₉₇₅. This mixture was sealed under high vacuum in a crucible and heated at 900 °C for 10 h, and subsequently cooled slowly to 600 °C in 4 days. Excess Te was separated using a high-temperature centrifuge, resulting in gold-colored, plate-like LaTe₃ crystals. The ARPES measurements presented here were performed at the SGM3 beamline at the ASTRID2 synchrotron facility[67]. ARPES data at SGM3 beamline were collected with an energy resolution of 18 meV at h$\nu$ = 24.4 eV. The angular resolution was 0.2° (0.008 Å⁻¹). The measurements were performed at 100 K with different photon energies ranging from 13 to 140 eV. Linearly polarized photon beam in the horizontal plane was incident at an angle of 50° with respect to the surface normal oriented along the analyzer axis. The analyzer slit is oriented vertically, and thus the detection plane is vertical. The polarization vector of the incident photon beam thus has two components parallel ($p$ polarization) and perpendicular ($s$ polarization) to the detection plane. The ARPES geometry is shown in Supplementary Fig. 26. All the measurements were done on freshly cleaved surfaces at a chamber base pressure better than $2 \times 10^{-10}$ mbar. The ARPES intensity plots measured as a function of photon energy (h$\nu$) have been converted to $k_y$ assuming the free electron final states[68]. The raw data ($E_{kin}$ vs h$\nu$) have been converted to $E$ vs $k_y$ by utilizing the expression $k_y = \frac{1}{\hbar}\sqrt{2m(E_{kin}cos^2\theta + V_0)}$, where $E_{kin}$ is the kinetic energy of the photoelectrons, $\theta$ is the emission angle and the inner potential $V_0$ has been assumed to be 7.5 eV based on the best matching with the ARPES data.

The curvature plots have been obtained as in our earlier work[69,70] with respect to both $E$ and $k$ axes following the method proposed by Zhang et al.[71]. This method improves the visibility of the weaker bands in the ARPES intensity plots and is better than the second derivative approach. The data analysis has been performed using IGOR pro (version 9). The dashed curves in Fig. 2e, f are obtained by a curve fitting the maxima of the MDCs. The maxima are identified by fitting the MDCs with Lorentzian functions. The least-square error method has been used for the curve fitting. The momentum resolution of ARPES represents the error in determining **q**$_{CDW}$.

The STM measurements were carried out at a base pressure of $2 \times 10^{-11}$ mbar using a variable temperature STM from Omicron Nanotechnology GmbH, while LEED and preliminary ARPES measurements were carried out in a workstation from Prevac sp. z o.o. STM was performed in the constant current mode using a tungsten tip that was cleaned by sputtering and voltage pulse method. The tip was biased and sample was kept at the ground potential. LEED is performed using a four grid rear view optics from OCI Vacuum Microengineering. Both STM and LEED were performed in the CDW state at room temperature. The error in the **q**$_{CDW}$ value from STM has been determined from the inverse of the step size of the topography image. In case of LEED, the error was determined from the pixel width of the pattern.

**Density functional theory.** The DFT calculations have been performed for the 7f and 29f structures of LaTe₃ with $C2cm$ space group ($SG$ #40). These structures have been derived from the experimental atomic positions at 100 K reported in ref. 33 using the PSEUDO program. The PSEUDO program[72] displaces the atoms to arrive at the 7f and 29f structures with $C2cm$ space group. The structure file for 29f LaTe₃ (Supplementary Data 1) is provided in the Supplementary Information. VESTA software has been used for Crystal structure visualization[73].

We have employed the DFT-based Vienna Ab-initio Simulation Package(VASP)[74,75] within the framework of the projector augmented wave method(PAW)[74,75] to investigate the electronic structure of LaTe₃. The exchange-correlation functional is treated under the generalized gradient approximation(GGA) given by Perdew, Burke, and Ernzerhof[76]. We have considered 11 valence electrons of the La atom ($5s^25p^65d^16s^2$) and 6 valence electrons of Te atom ($5s^25p^4$) in the PAW pseudopotential. The energy cut-off is set to 500 eV for the expansion of the planewaves. The convergence criterion for energy in the self-

consistent-field cycle and total force tolerance on each atom are taken to be $10^{-6}$ eV and 0.02 eV/Å, respectively. The SOC is employed by a second-variation method as implemented in the VASP code[75]. To calculate the EBS, we have unfolded the band structure of the CDW state into the primitive BZ of the non-CDW state, using the PyProcar python code[58]. All the DFT bands (and consequently the EBS) are rigidly shifted to larger $E$ by 0.1 eV with respect to the $E_F$ for comparison with the ARPES data.

The VASP calculations were carried out with the experiment-based 7f and 29f structures without geometry optimization. This approach has been used in the literature for complicated structures with a large unit cell, especially with modulation or anti-site defects[27,77–80]. Nevertheless, our calculation with full geometry optimization including the van der Waals interaction by the DFT-D3 method (the electron-phonon interaction was not considered) retains the CDW state with $\mathbf{q}_{CDW}$ practically unchanged. But, the CDW amplitude decreases drastically by 80% compared to the experimental value from x-ray crystallography. Also, the CDW gap along $\Gamma Z$ is not obtained, as shown in Supplementary Fig. 27a. The EBS calculation shows that the spectral weight of the shadow bands is much lower and so these are virtually non-existent (Supplementary Fig. 27b). These results are in stark disagreement with ARPES and justifies the use of the experiment-based structure without optimization for performing the DFT calculations.

In order to reconfirm the band crossings and to identify their irreps, DFT calculations have been performed using the all-electron WIEN2k programme package[81] and Quantum Espresso software package[82]. Supplementary Fig. 28 shows that the bands are in agreement between VASP and WIEN2k. The latter was performed with energy cut-off of 16 Ry, where the $R_{MT}K_{max}$ value is taken to be 9.5. Further, we have used 10 for the maximum value of angular momentum for the (l,m) expansion of wave function or density. Convergence criteria for energy and charge have been taken to be $10^{-5}$ Ry and 0.001 $e^-$, respectively. Quantum Espresso calculations have been carried out using fully relativistic PAW pseudopotentials for La and Te atoms. A planewave cutoff of 80 Ry and a $6 \times 6 \times 1$ $\mathbf{k}$-grid were taken along with the energy accuracy of $10^{-8}$ Ry.

## Data availability
The data that support the findings of this study are available from the corresponding authors upon request.

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

## Acknowledgements

S.S., P.S., and S.R.B. gratefully acknowledge the financial support from the Department of Science and Technology, Government of India within the framework of the DST-Synchrotron-Neutron Project to perform experiments at ASTRID2 synchrotron facility. A part of this work was supported by VILLUM FONDEN via the Centre of Excellence for Dirac Materials (Grant No. 11744). Anushree Roy is thanked for providing the Raman data. The Computer division of Raja Ramanna Centre for Advanced Technology is thanked for installing the DFT codes and providing support throughout.

## Author contributions

S.S., P.S., D.C., and S.R.B. conducted the ARPES measurements with assistance and support from M.B. and P.H. LEED was carried out by S.S. and P.S., while STM was performed by V.K.S. J.B. and R.D. did the DFT calculations under the supervision of A.C. The explanation of the results was provided by S.S., T.D., A.C., and S.R.B. The single crystals of LaTe$_3$ were grown by A.P., S.R., and P.M., the latter introduced us to this system. S.S. analyzed the experimental data with initial help from D.C., performed the post-analysis of the DFT results with some inputs from J.B., and prepared the figures. The project was planned and led by S.R.B. who wrote the paper with significant contributions from S.S., P.H., and A.C.

## Competing interests

The authors declare no competing interests.
