## [Peer Review File · Nature Communications]

Reviewers' Comments:

Reviewer #1:

Remarks to the Author:

The paper by S. Sarkar et al. presents systematic high-resolution ARPES measurements and first-principle calculations of the electronic structure of LaTe₃, a well-studied quasi-2D charge density wave (CDW) material. Corroborating with previous XRD and ARPES studies, the current paper confirmed (1) a unidirectional CDW modulation along the c axis with a q vector of $\sim 2/7 c^*$, (2) the crossings of main and CDW shadow bands, which form a Dirac nodal line in the absence of SOC. Importantly, the authors claim the existence of a symmetry-enforced KNL when SOC is considered. I have the following comments/suggestions to improve the paper:

(1) My major concern, and it is a great concern, is the basis of the paper, i.e., the symmetry of LaTe₃. It's well established that the rare-earth tritellurides (RTe₃) host an incommensurate CDW with a q vector of, approximately but not exactly, $2/7 c^*$. Strictly speaking, incommensurate CDW has no translational symmetry. That's why one can only define an average crystal structure and space group for LaTe₃ [37]. Similarly, DFT can only approximate the electronic structure with a commensurate periodicity, i.e., (1x1x7) supercell in LaTe₃. How can one justify the existence of symmetry-enforced KNL in an incommensurate CDW?

(2) On the other hand, assuming LaTe₃ hosts a commensurate CDW that belongs to C2cm (SG #40), as claimed in the paper. However, according to the theoretical prediction, i.e., table 1 of Ref. [25], space group 40 does not host KNL. Am I missing something?

(3) The authors should also be more cautious in their statements. For example, the author state that "While there are no DFT investigations, angle-resolved photoemission spectroscopy (ARPES) measurements on LaTe₃ are also scarce in literature." Actually, there are already published DFT results on LaTe₃ [<https://doi.org/10.1016/j.physb.2022.413988>]. I strongly recommend the authors conduct comprehensive literature research on rare-earth tritellurides and KNLs.

(4) Fig. 3h and o are misleading, as the nodal line is highly-dispersive along the k_x direction.

(5) Fig. 3(a) and 3(i) show two flat bands, what's their origin? Why they are absent in the ARPES measurements?

Reviewer #2:

Remarks to the Author:

Sarkar et al. study the electronic structure of the layered charge-density (CDW) material LaTe₃ using ab-initio band theory calculations (DFT) and angle-resolved photoemission spectroscopy (ARPES). In comparison to previous APRES on rare earth tritellurides, their data are of unprecedented quality, revealing band crossing of shadow bands reflected at the boundary of the reduced Brillouin zone in the charge-ordered state, which they identify as Kramers nodal lines (KNL). The analysis and discussion are heavily based on the space group symmetry of the charge-ordered state as derived from a previous x-ray scattering work (Ref. 37).

I support the publication of this manuscript in Nature Communications after revisions.

1. The reported ordering vector of the CDW state is incommensurate in previous work, based on high-resolution x-ray diffraction (e.g. Ref. 37). The authors present scanning-tunneling microscopy and LEED data claiming the formation of a commensurate superstructure with $q = 2/7 c^*$. Although the authors assert their unit cell is consistent with previous work, Ref. 37 gives $q = 0.2719(3) c^*$, in discrepancy with the authors' claim of $q = 2/7 = 0.2857 c^*$. At a minimum the authors have to discuss the inconsistency and add a section to the SI explaining why a slight discommensuration (or true incommensurability) does not affect their conclusions, which are based on DFT calculations in a

commensurate unit cell.

2. The C2cm structure reported in Ref. 37 is non-centrosymmetric, with polar a-axis. As the previous reference (Ref. 37) is also not very clear about this point, please explain in much more detail how the polar distortion comes to pass in the CDW state; please clarify (in text or illustrations) how the atoms are displaced from their positions in the original Cmc structure, and how this distortion realizes the polar state. Perhaps related to this question, the caption of Fig. S1 is extremely unclear; please expand.

3. For publication in a leading journal, the polar (inversion breaking) nature of the charge-ordered structure should be confirmed by an independent experimental probe, such as second harmonic generation of light or nonlinear electric transport. Please add supporting experimental evidence to the SI or main text. The C2cm structure and its noncentrosymmetric nature are the key underpinning of the authors' theoretical analysis and should be verified experimentally. It is not sufficient to lean on the previous diffraction work, in the eyes of this referee.

4. In Fig. S2, the authors deduce the ordering vector of the CDW from relatively broad intensity maxima in the Fourier transform of their STM data, and from LEED intensity. They claim on page 4 of the SI that "qCDW to be $0.275 \pm 0.001 c^*$ "; it is very hard to see from the very broad data in Fig. S2(d) how this high resolution of one milli-r.l.u. could be realized.

5. Minor points: In Fig. 2 h-k, what is the intensity scale or dot size representing? In the caption of Fig. 3, a large amount of jargon is thrown at the reader. Please make this more easily readable, especially as regards the irreg. language. In Fig. 2f,g the dots are hard to see; also, is it not better to label $\Gamma_2 \rightarrow X$ above these panels, as in Fig. 2(n)? In Fig. 4(a), the color shading around X (orange and green) is not explained in the caption. The x-axis of panel (c) should have the same scale as (b). Please explain in the caption of S18 what the different conditions or assumptions in the two calculations were, even if this is roughly indicated in Methods. It is not sufficient to write "by VASP" or "by WIEN2k".

Reviewer #3:

Remarks to the Author:

The interplay between topology and correlated electronic states is an important issue in the studies of topological materials and remains largely unexplored experimentally. Focusing on the band structure topology of LaTe3, a noncentrosymmetric material with time-reversal symmetry and CDW at a high transition temperature of 670 K, Shuvam Sarkar et al. mostly present ARPES data, DFT and band unfolding EBS calculations, as well as symmetry discussions and claim that LaTe3 hosts a Kramers nodal line (KNL). The ARPES data of cleaved bulk LaTe3 is compared with the calculated results, and is found to show certain consistency with the presence of the KNL. DFT calculations include SOC reveals that the KNL imposes gapless crossings between the bilayer-split CDW-induced shadow bands and the main bands, while ARPES data corroborate the presence of the KNL and show that the crossings traverse the Fermi level. Though the RTe3 material system could be potentially very intriguing I feel that this work needs major revision.

Main comments

1. The title "Charge density wave induced nodal lines in LaTe3" is to some extent misleading. What is the exact relationship between the CDW state and bilayer-splitting? This might be the chicken or the egg problem. Would there be more fundamental reasons such as surface reconstruction, genuine termination effects, or effects due to dimensionality and/or confinement? The statement "Since the splittings are visible in both the states, these are not related to the CDW, but play an interesting role in giving rise to two types of nodal lines in LaTe3." in Supplementary Note 2 also confuses me. The author should try to clarify this in the manuscript.

2. Considering the recent ARPES results of NdTe3 (arXiv: 2209.04226), especially, Fig.2 (b) inside, where the bilayer splitting is not observed, a natural question can be raised, say, what is the situation for LaTe3? I recommend the authors add such details at least in the Supplementary information.

3. Although overall the theoretical calculations match the ARPES data, I struggle to distinguish the bilayer splitting of the shadow bands in Fig.2 (a) and assume that the splitting tendency is small near EF, while larger at around 0.8 eV, which is in contrast to the calculations in Fig.2 (h). Besides, the splitting features near EF haven't been well resolved in the data. Since these band features seem to be quite close to EF both experimentally and theoretically, have the authors tried potassium surface doping of the system to gain the features above EF? And are there any attempts to produce few-layer material? This may also help us understand the mechanism behind the bilayer-splitting.

4. The nodal line related data presented in the manuscript is abstract and sometimes hard to follow. I understand there is energy dispersion along the k_x direction for the nodal lines, still it would be relatively easy for us to understand if the authors present the energy contours of the k_x - k_z plane.

5. The authors do mention the orbital characteristics of the bands and show the p-orbital characters from the EBS calculations in Fig. S12. However, some essential information on the ARPES experiment is missing. What is the photon polarization used, and what is the measurement scheme of the polarization-dependent ARPES? I recommend the authors add the information wherever appropriate.

6. The ab initio DFT calculation is based on a realistic experiment-based structure. Usually, the band structure topology is very sensitive to the crystal constants, have the authors performed calculations using just the theoretical self-consistent crystal structure? These calculated results should be added to the Supplementary information as reference.

Additional comments

7. Such as the bottom axis of Fig.3 (m), which is not consistent with Fig.3 (l), the authors should correct these inconsistencies since the band crossings should locate at the same k_z point.

8. Also in the caption of Fig.3, the authors mentioned that "The vertical dashed line represents the k_z point on Γ_2X_2 i.e., the Σ line." I checked that in Fig.1 (d) where the " Σ " label is with the gray arrow pointing at the Γ -X line. Is this a typo? A highlighted guiding line connecting the high-symmetry points in 3D would be also very helpful.

In conclusion, this manuscript deserves further consideration. I would suggest the authors to do this sort of study and revision before resubmission as it is crucial for interpreting both the ARPES and calculated results.

REPLY TO THE REVIEWER COMMENTS

Reply to the comments of Reviewer #1:

Comments: *The paper by S. Sarkar et al. presents systematic high-resolution ARPES measurements and first-principle calculations of the electronic structure of LaTe₃, a well-studied quasi-2D charge density wave (CDW) material. Corroborating with previous XRD and ARPES studies, the current paper confirmed (1) a unidirectional CDW modulation along the c axis with a q vector of $\sim 2/7 c^*$, (2) the crossings of main and CDW shadow bands, which form a Dirac nodal line in the absence of SOC. Importantly, the authors claim the existence of a symmetry-enforced KNL when SOC is considered. I have the following comments/suggestions to improve the paper:*

Reply: We appreciate the Reviewer's thorough examination of our submitted manuscript. We have addressed all the criticisms and questions of the Reviewer, as outlined below. In the process, and especially with the addition of new results and insights obtained from DFT calculations and experimental data, we feel the paper has been substantially improved. We thank the Reviewer for that. Please note that when referring to the figures, references, page numbers, and sections, we refer to the original version as R0 and the revised version as R1 for clarity. The R1 version's textual modifications are displayed in blue color.

Comment: *(1) My major concern, and it is a great concern, is the basis of the paper, i.e., the symmetry of LaTe₃. It's well established that the rare-earth tritellurides (RTe₃) host an incommensurate CDW with a q vector of, approximately but not exactly, $2/7 c^*$. Strictly speaking, incommensurate CDW has no translational symmetry. That's why one can only define an average crystal structure and space group for LaTe₃ [37]. Similarly, DFT can only approximate the electronic structure with a commensurate periodicity, i.e., $(1 \times 1 \times 7)$ supercell in LaTe₃. How can one justify the existence of symmetry-enforced KNL in an incommensurate CDW?*

Reply: We concur with the Reviewer that this is an important point. For this reason, this point was discussed in section III of the R0 version on page 17, but we accept that it was a short discussion that perhaps did not convey the message clearly and was not established by DFT calculation. So, in the R1 version, we elaborate on this and perform new DFT calculations with a much larger unit cell $(1 \times 1 \times 29)$ representing incommensurate LaTe₃ within the experimental accuracy of x-ray crystallography to establish the existence of the KNL.

It is well known that DFT requires a translationally periodic system and so in the R0 version we considered a commensurate structure with $q_{\text{CDW}} = \frac{2}{7}c^* = 0.2857c^*$, rounded up to the 4th decimal place. Importantly, it has the same basic space group (C2cm) as determined experimentally. However, as the Reviewer has pointed out, this q_{CDW} value is different from

the incommensurate value of $q_{\text{CDW}} = 0.2757(4)c^*$ reported by x-ray crystallography. The latter value is the most precise value of q_{CDW} reported in the literature so far for LaTe_3 .

It may be noted that a legitimate way of calculating the band structure of incommensurate systems is described in *International Tables for Crystallography* Vol. C, Chapter 9.8, page 907 [R1:Ref. 41] by Janssen, Janner, Looijenga-Vos, and de Wolff that states “*There is more than one way for expressing the long-range order present in an incommensurate crystal in terms of symmetry. One natural way is to adopt the point of view that the measuring process limits the precision in the determination of the modulation wavevector. Accordingly, one can try an approximation of the modulation wavevector q by a commensurate one: an irrational number can be approximated arbitrarily well by a rational one.*” This is also mentioned in the well referred book on incommensurate crystallography by van Smaalen [R1:Ref.42]: “*Whether the components of modulation wave vectors are rational or irrational numbers is only of secondary importance, because an experiment can never distinguish between an irrational number and some rational approximant.*” Thus, an incommensurate structure can be represented as a large unit cell commensurate structure such that q_{CDW} is the same as the incommensurate value up to the limit of experimental accuracy. This approach would be always valid because no experiment would ever have the precision to definitively state that q_{CDW} is irrational in the strict mathematical sense because a number with finite decimal digits can always be expressed as a ratio of two integers. Indeed, even within inorganic materials, there are translationally periodic systems with very large unit cells [R1: Ref. 43,44,45]. Moreover, in the literature, LaTe_3 (and the whole RTe_3 series) has been dubbed as “incommensurate” since its q_{CDW} value deviates from the small integer ratio. This was acknowledged by Fang et al. [PRL **99**, 046401 (2007)] who showed the incommensurate nature of q_{CDW} in TbTe_3 as “... *it is difficult to distinguish incommensurate against a high denominator fraction commensurate.*”

Based on the above discussed approach for incommensurate systems, we considered different possible combinations of the numerator and the denominator of the fraction that could represent the q_{CDW} from x-ray crystallography. We found that a 29-fold (29f) structure with $q_{\text{CDW}} = \frac{8}{29}c^* = 0.2759c^*$ matches $q_{\text{CDW}} = 0.2757(4)c^*$ within the experimental accuracy. Importantly, the 29f structure has the same C2cm space group (see Methods) and thus the same symmetries as the 7f structure, as was briefly mentioned in the R0 version. There are 232 atoms in its unit cell with positions almost coinciding with those from Ref. 33 [230 atoms have zero (0.0000 Å) displacement, while only 2 atoms show a displacement of 0.0001Å]. The cif file (file name: 29fLaTe3.cif) of the 29f structure is provided in the Supplementary Information of the R1 version that for all practical purposes i.e. within the experimental accuracy of x-ray crystallography can be considered as the actual structure of LaTe_3 .

In the R1 version, costly DFT calculations were performed for the 29f structure. Its unit cell is shown in R1:Fig.S1. Moreover, the equivalence of 29f structure with that from x-ray crystallography is illustrated in R1:Fig. S2, where the CDW modulation wave and the Te atom for the 29f structure are indistinguishable from Ref. 33. The origin of noncentrosymmetry and the existence of other symmetries are shown in R1:Fig.S3. These discussions appear at the beginning of the Results section in the 2nd and 3rd paragraph of R1:page 4 and continue to the 1st and partly 2nd paragraph of R1:page 6.

The band structure results that demonstrate the existence of the KNL and the spinless nodal line in the 29f structure are presented in the Discussion (section III) on R1:page 17-18. Since the 29f results are important, a new figure (R1:Fig.5) is included in the main manuscript to show the band structure results, while Figs. S24 and S25 appear in Supplementary Information (please see the “Summary of Major changes” at the end of this Reply).

To conclude, in the R1 version through explicit DFT calculation for the 29f structure, we establish the KNL in LaTe₃. A qualitative improvement in our work motivated by the Reviewer’s comments (and also a similar Comment #1 by Reviewer #2, please see our reply to that comment also) is that, to the best of our knowledge, it is the first time that DFT has been performed for an incommensurate CDW system with its actual structure within the experimental accuracy.

Comment: (2) *On the other hand, assuming LaTe₃ hosts a commensurate CDW that belongs to C2cm (SG #40), as claimed in the paper. However, according to the theoretical prediction, i.e., table 1 of Ref. [25], space group 40 does not host KNL. Am I missing something?*

Reply: Kindly note that Table 1 of R1:Ref. 25 lists with only the symmorphic space groups. This is clearly stated in the caption of Table 1.

In contrast, the C2cm space group of LaTe₃ is nonsymmorphic. For nonsymmorphic space groups, R1:Ref. 25 states that *“Indeed, KNLs also appear in all crystals that are noncentrosymmetric and nonsymmorphic. Particularly, there are always KNLs coming out of the Γ points of nonsymmorphic crystals.”* Indeed, in our case, the KNL (Σ) line starts from the Γ point. Furthermore, in Supplementary Note 2 of R1:Ref. 25, it is proved how the C2 symmetry along k_x , in addition to time reversal and mirror symmetry constrains the KNL along k_x , as in the present case. Supplementary Note 4 in R1:Ref.25 determines the KNL from compatibility relations.

Comment: (3) *The authors should also be more cautious in their statements. For example, the author state that “While there are no DFT investigations, angle-resolved photoemission spectroscopy (ARPES) measurements on LaTe₃ are also scarce in literature.” Actually, there are already published DFT results on LaTe₃ [<https://doi.org/10.1016/j.physb.2022.413988>]. I strongly recommend the authors conduct comprehensive literature research on rare-earth tritellurides and KNLs.*

Reply: We apologize for this omission. We have referred to this important DFT work at the end of the 2nd paragraph R1:page 3 as R1:Ref. 37 with the inclusion of the following sentence “A recent DFT study on a free-standing monolayer of LaTe₃ revealed that tensile strain would increase the CDW order, while compressive strain would suppress it, and superconductivity could develop. [37].” We have also performed a literature survey as the Reviewer suggested.

Comment: (4) *Fig. 3h and o are misleading, as the nodal line is highly-dispersive along the k_x direction.*

Reply: We have included the dispersion in the modified Figs.3h and 3o in the R1 version. Moreover, the last sentence of the figure caption of the R0 version is deleted.

Comment: (5) *Fig. 3(a) and 3(i) show two flat bands, what’s their origin? Why they are absent in the ARPES measurements?*

Reply: The two flat bands result from the band folding in the modulated structure. As a result of their reduced spectral weight, these are not visible in the unfolded effective band structure (EBS). In the R1 version, we have provided a figure in the Supplementary Information (Fig. S18) to illustrate this. As ARPES intensity plots are the representation of the EBS, the flat bands are not detected in ARPES either. A part of this discussion is in the 2nd paragraph of R1:page 11.

(please turn over)

Reply to the comments of Reviewer #2:

Comment: Sarkar et al. study the electronic structure of the layered charge-density (CDW) material LaTe₃ using ab-initio band theory calculations (DFT) and angle-resolved photoemission spectroscopy (ARPES). In comparison to previous APRES on rare earth tritellurides, their data are of unprecedented quality, revealing band crossing of shadow bands reflected at the boundary of the reduced Brillouin zone in the charge-ordered state, which they identify as Kramers nodal lines (KNL). The analysis and discussion are heavily based on the space group symmetry of the charge-ordered state as derived from a previous x-ray scattering work (Ref. 37).

I support the publication of this manuscript in Nature Communications after revisions.

Reply: We welcome the Reviewer's insightful comments and the appreciation of our work. We have revised the manuscript extensively by adding new theoretical results and experimental data. We have also improved the presentation (please see the Summary of Changes at the end of our Reply). Thus, we have addressed all the questions and suggestions for the Reviewer. This added to the quality of our manuscript, and we are grateful. Kindly note that when referring to figures, references, page numbers, and sections, we refer to the original version as R0 and the amended version as R1 for clarity. The R1 version's textual modifications are displayed in blue color.

Comment: 1. The reported ordering vector of the CDW state is incommensurate in previous work, based on high-resolution x-ray diffraction (e.g. Ref. 37). The authors present scanning-tunneling microscopy and LEED data claiming the formation of a commensurate superstructure with $q = 2/7 c^*$. Although the authors assert their unit cell is consistent with previous work, Ref. 37 gives $q = 0.2719(3) c^*$, in discrepancy with the authors' claim of $q = 2/7 = 0.2857 c^*$. At a minimum the authors have to discuss the inconsistency and add a section to the SI explaining why a slight discommensuration (or true incommensurability) does not affect their conclusions, which are based on DFT calculations in a commensurate unit cell.

Reply: The comment is important and well taken. We agree with the Reviewer that it should be clarified whether the incommensurate structure with $q_{\text{CDW}} = 0.2757(4)c^*$ and its 7-fold (7f) commensurate representation with an approximate $q_{\text{CDW}} = \frac{2}{7} c^*$ provides similar conclusions. In the R1 version, we have now elaborated on our earlier discussion in R0:page17 about this in subsection IIA in R1:page 4 where a 29 fold (29f) structure is presented (see also Figs. S1-2) that represents the actual structure of incommensurate LaTe₃ within the experimental accuracy. The results of the new DFT calculations with the 29f structure are presented in the Discussion section (section III) on R1:pages 17-18. A similar comment has been made by Reviewer #1, so please also see our reply to Comment #1 of Reviewer #1, although our Replies to both Reviewers are mostly similar.

In the R0 version, we have performed the DFT calculations with a 7f structure with $q_{\text{CDW}} = \frac{2}{7}c^*$ = 0.2857 c^* . On the other hand, the incommensurate value of q_{CDW} is 0.2757(4) c^* at 100 K (note here: our ARPES measurements were carried out at this temperature). This value [0.2757(4) c^*] is stated in Table 1S and Table 2S of R1:Ref.33. This is the most accurate determination of q_{CDW} for LaTe₃ that exists in literature until the date at this temperature.

We are on the same page with the Reviewer's comment that based on $q_{\text{CDW}} = 0.28c^*$ determined from our ARPES, STM, and LEED data, it should not be claimed that LaTe₃ has commensurate $q_{\text{CDW}} = \frac{2}{7}c^* = 0.2857c^*$. This is because the errors for determining q_{CDW} by the above-mentioned methods are in the range of 0.005-0.02 c^* that is considerably larger compared to x-ray crystallography. So, the last sentence of the 1st paragraph of R0:page 6 has been modified in the 2nd paragraph of R1:page7 to " *$q_{\text{CDW}} = 0.28 \pm 0.005c^*$ determined in this way agrees with the values obtained from STM and LEED, as well as that from transmission electron microscopy ($q_{\text{CDW}} = 0.28 \pm 0.01c^*$). The inaccuracy in these approaches for calculating q_{CDW} is substantially larger than the error in x-ray crystallography, which is 0.0004 c^* [R1:Ref.33].*"

Our approach for performing DFT for the incommensurate structure of LaTe₃ is based on the following. (i) An article in *International Tables for Crystallography* Vol. C, Chapter 9.8, page 907 [R1:Ref.41] by Janssen *et al.* states "*There is more than one way for expressing the long-range order present in an incommensurate crystal in terms of symmetry. One natural way is to adopt the point of view that the measuring process limits the precision in the determination of the modulation wavevector. Accordingly, one can try an approximation of the modulation wavevector q by a commensurate one: an irrational number can be approximated arbitrarily well by a rational one.*" (ii) A standard book on incommensurate crystallography by van Smaalen [R1: Ref. 42] that states "*Whether the components of modulation wave vectors are rational or irrational numbers is only of secondary importance, because an experiment can never distinguish between an irrational number and some rational approximant.*"

Thus, from the above-mentioned discussion, it is clear that an incommensurate structure can be represented as a commensurate structure with a large unit cell such that q_{CDW} is the same as the incommensurate value within the experimental accuracy. This approach would be always valid because no experiment would ever have the precision to definitively state that q_{CDW} is irrational in the strict mathematical sense because a number with finite decimal digits can always be expressed as a ratio of two integers that is the definition of a rational number.

Based on the above approach, we considered different possible combinations of the numerator and the denominator of the fraction that could represent q_{CDW} . We found that a 29-

fold (29f) structure with $q_{\text{CDW}} = \frac{8}{29} c^* = 0.2759 c^*$ exactly matches $q_{\text{CDW}} = 0.2757(4)c^*$ within the experimental accuracy. Importantly, the 29f structure has the same C2cm space group and thus the same symmetries as the 7f structure. There are 232 atoms in the unit cell of the 29f structure with positions almost coinciding with those from Ref. 33 [230 atoms have zero (0.0000 Å) displacement, while only 2 atoms show a displacement of 0.0001 Å]. The cif file (file name: 29fLaTe3.cif) of the 29f structure is provided in the SI in the R1 version that for all practical purposes represents the actual structure of LaTe₃. Here, we would like to express our gratitude to the authors of Ref. 33 for making their cif file available in the public domain without which the present DFT calculations would not have been possible.

In the R1 version, we performed difficult and expensive DFT calculations for the 29f structure with a 232-atom large unit cell. R1:pages 17-18 (section III) discuss the band structure studies that show the KNL and spinless nodal line in the 29f structure (see R1:Fig.5 and R1:Figs.S24 and S25). All the major changes in the manuscript are summarized under “Summary of Major changes” at the end of this Reply.

We also noted that the q_{CDW} value we have taken above [$0.2757(4)c^*$] is slightly different from that mentioned by the Reviewer [$0.2719(3)$]. We had noticed that the latter value is also mentioned in Ref. 33 as q_{CDW} of LaTe₃ at 100 K. So, to resolve this confusing situation and decipher which is the correct value in Ref. 33 (one or the other is a printing error), we considered the cif file (available at <https://ndownloader.figstatic.com/files/4761340>). In the cif file, the value of q_{CDW} is mentioned as $0.2757(4)c^*$. Moreover, to confirm this, in R1:Fig.S2 we plotted the positions of the Te atoms from the coordinates provided in this cif file and fitted the modulation with a sinusoidal curve. We find the correct q_{CDW} value to be $0.2757c^*$.

In conclusion, we demonstrate the existence of the nodal lines in LaTe₃ in the R1 version using the novel DFT calculations for the 29f structure. This was a daunting task that we could barely manage to carry out with our computational resources. To the best of our knowledge, this is the first time DFT has been performed for an incommensurate CDW system with its actual structure within the experimental accuracy of x-ray crystallography.

Comment: 2. The C2cm structure reported in Ref. 37 is non-centrosymmetric, with polar a-axis. As the previous reference (Ref. 37) is also not very clear about this point, please explain in much more detail how the polar distortion comes to pass in the CDW state; please clarify (in text or illustrations) how the atoms are displaced from their positions in the original Cmcm structure, and how this distortion realizes the polar state. Perhaps related to this question, the caption of Fig. S1 is extremely unclear; please expand.

Reply: Following Reviewer’s suggestion, we have added two new figures (R1:Figs.S2 and S3 to replace R0:Fig.S1.

In R1:Fig.S2, we directly compare the Te (Te2 and Te3) atoms' positions between R1:Ref. 33 and the 29f and 7f structures in panels **a** and **b**, respectively. These are also compared with the undistorted positions in the original non-CDW Cmcm structure. Thus, in accordance with the Reviewer's question, here we graphically show "*how the atoms are displaced from their positions in the original Cmcm structure*".

Next, from R1:Fig.S3a we show how the noncentrosymmetry arises: the distortion along the x direction (δx) breaks the M_x mirror symmetry that was present in the non-CDW structure (the M_x mirror can be visualized to be perpendicular to the plane of the paper along the dashed line).

We further show the other symmetries in the CDW state that determine the polar axis. For this, we consider not only the δx distortion (Fig.S3a), but also the δy , and δz distortions although these are order of magnitude smaller (Fig.S3b,c). The green vertical line in these panels shows that the M_z mirror is preserved. Fig.S3d shows that the glide symmetry (M_y mirror and $c/2$ translation) is also preserved in the CDW state. The intersection of M_z and M_y gives the polar axis (red arrow in Fig, S3e) since it has to be parallel to both the M_z mirror in the y-x plane and the M_y glide mirror in the x-z plane.

Comment: 3. For publication in a leading journal, the polar (inversion breaking) nature of the charge-ordered structure should be confirmed by an independent experimental probe, such as second harmonic generation of light or nonlinear electric transport. Please add supporting experimental evidence to the SI or main text. The C2cm structure and its noncentrosymmetric nature are the key underpinning of the authors' theoretical analysis and should be verified experimentally. It is not sufficient to lean on the previous diffraction work, in the eyes of this referee.

Reply: X-ray crystallography is a standard technique to probe of the crystal structure. However, we agree with the Reviewer's comment that "*an independent experimental probe*" other than the previous x-ray diffraction work (R1:Ref. 33) will provide additional evidence of noncentrosymmetry.

One such experimental probe is Raman vibrational spectroscopy which has been widely used as a definitive proof of noncentrosymmetry. For example, Zhang et al. identified five lattice vibrational modes that are Raman-active only in the low-temperature noncentrosymmetric structure for MoTe_2 [Ref. 86]. Raman spectroscopy was also used to establish noncentrosymmetry in TaIrTe_4 , a type-II Weyl fermion [Liu et al., *Adv. Mater.* 2018, 1706402]. Lavagnini et al. (R1: Ref. 34) performed Raman spectroscopy for a series of $R\text{Te}_3$ compounds including LaTe_3 . LaTe_3 was studied in the CDW phase at room temperature. Utilizing a first-principles calculation considering a $(1 \times 1 \times 7)$ supercell for the CDW state, noncentrosymmetric vibrational modes with B_1 symmetry were identified in LaTe_3 [Ref.34].

In the R1 version, we provide results of our Raman vibrational spectroscopy measurement on LaTe_3 in the CDW phase at room temperature in R1: Fig. S5. The experiment was performed with a 532 nm laser source operating at 0.5 milliwatt. The spectrum is in agreement with those from literature (Refs. 34 and 87) that are shown staggered along the vertical axis for comparison. The P2 and P4 peaks (highlighted by the gray rectangles) contain the B1 symmetry mode, which is an irreducible representation of the C_{2v} point group and is a direct signature of noncentrosymmetry in LaTe_3 .

Furthermore, inversion symmetry breaking is also portrayed by our STM measurement (Fig. 1b), where a distorted Te net is observed (in contrast to a square net for an undistorted structure). Also the dashed white lines (included in R1:Fig.1b) show that the average positions of the adjacent Te chains are not equidistant. These observations show that the M_x mirror symmetry is broken, leading to noncentrosymmetry. We agree that SHG and nonlinear electric transport measurements are standard techniques for probing symmetry breaking in solids. As far as we know, such studies on LaTe_3 have not been performed. So, it would be interesting to pursue further research on LaTe_3 using these techniques.

Thus, besides x-ray crystallography, in the R1 version, we provide experimental evidence of noncentrosymmetry from two other independent experimental probes such as Raman vibrational spectroscopy and STM. This is mentioned in the 2nd paragraph of R1:page 6 and R1:Ref.34 is included.

Comment: 4. In Fig. S2, the authors deduce the ordering vector of the CDW from relatively broad intensity maxima in the Fourier transform of their STM data, and from LEED intensity. They claim on page 4 of the SI that “ q_{CDW} to be $0.275 \pm 0.001 c^*$ ”; it is very hard to see from the very broad data in Fig. S2(d) how this high resolution of one milli-r.l.u. could be realized.

Reply: We apologize for this mistake. The error turns out to be $0.012c^* \approx 0.01c^*$ and we have rectified the value of q_{CDW} from LEED to be $0.28 \pm 0.01c^*$ in the 2nd paragraph of R1:page 4 of the Supplementary Information.

Comment: 5. Minor points: In Fig. 2 h-k, what is the intensity scale or dot size representing?

Reply: The intensity scale represents the spectral weights calculated from band unfolding. The size of the dots is related to the instrumental resolution. This is now mentioned in the caption of Fig. 2.

Comment: In the caption of Fig. 3, a large amount of jargon is thrown at the reader. Please make this more easily readable, especially as regards the irrep. language.

Reply: To simplify the caption, we have removed unnecessary definitions (already defined in the text) such as: “ $\mu \otimes sd$ ”, “ $md \otimes su$ ”, “ $\mu \otimes su$ ”, “ $md \otimes sd$ ”, and “irreps”. Furthermore, we have simplified and shortened the caption. Also, “KNL” has been defined in the title of the figure caption.

Comment: *In Fig. 2f,g the dots are hard to see; also, is it not better to label Gamma2 -> X above these panels, as in Fig. 2(n)?*

Reply: We think the Reviewer means Fig. 3f,g. We have improved the visibility of the dots in these figures by making them bigger. In Figs. 3f,g, the k_z values where the cuts are taken along k_x are now mentioned at the top.

Comment: *In Fig. 4(a), the color shading around X (orange and green) is not explained in the caption. The x-axis of panel (c) should have the same scale as (b).*

Reply: We have explained this in the caption of Fig. 4 in the R1 version. Also, the x-axes of all three panels (a-c) now have the same scaling to avoid any confusion.

Comment: *Please explain in the caption of S18 what the different conditions or assumptions in the two calculations were, even if this is roughly indicated in Methods. It is not sufficient to write “by VASP” or “by WIEN2k”.*

Reply: In the R1 version, we have modified the caption of R0:Fig. S18 (R1:Fig.S28) to include a short description of the two different methods of DFT calculation. The Methods section is referred to for details.

(please turn over)

Reply to the comments of Reviewer #3:

Comment: *The interplay between topology and correlated electronic states is an important issue in the studies of topological materials and remains largely unexplored experimentally. Focusing on the band structure topology of LaTe₃, a noncentrosymmetric material with time-reversal symmetry and CDW at a high transition temperature of 670 K, Shuvam Sarkar et al. mostly present ARPES data, DFT and band unfolding EBS calculations, as well as symmetry discussions and claim that LaTe₃ hosts a Kramers nodal line (KNL). The ARPES data of cleaved bulk LaTe₃ is compared with the calculated results, and is found to show certain consistency with the presence of the KNL. DFT calculations include SOC reveals that the KNL imposes gapless crossings between the bilayer-split CDW-induced shadow bands and the main bands, while ARPES data corroborate the presence of the KNL and show that the crossings traverse the Fermi level. Though the RTe₃ material system could be potentially very intriguing I feel that this work needs major revision.*

Reply: We appreciate the Reviewer's positive comments on our manuscript. We have significantly revised the manuscript, addressing all the Reviewer's suggestions and questions. The results of new theoretical calculations and experimental research, as explained in detail below, have been included. Please note that when referring to the figures, references, page numbers, and sections, we refer to the original version as R0 and the amended version as R1 for clarity. The R1 version's textual modifications are displayed in blue color.

Comment: *1. The title "Charge density wave induced nodal lines in LaTe₃" is to some extent misleading. What is the exact relationship between the CDW state and bilayer-splitting? This might be the chicken or the egg problem. Would there be more fundamental reasons such as surface reconstruction, genuine termination effects, or effects due to dimensionality and/or confinement? The statement "Since the splittings are visible in both the states, these are not related to the CDW, but play an interesting role in giving rise to two types of nodal lines in LaTe₃." in Supplementary Note 2 also confuses me. The author should try to clarify this in the manuscript.*

Reply: In order to begin providing an answer to the Reviewer's query regarding the bilayer-splitting and its connection to the CDW state, we perform new DFT calculations. If the bilayer splitting decreases with the increase of the Te-Te interlayer separation, this will prove that it is related to the Te bilayer coupling. So, we increase the separation from 3.71Å by 5% and 10% in the y direction keeping the rest of the interatomic distances - for example within the La-Te1 block- unchanged in the non-CDW state of LaTe₃. R1: Fig.S8 shows that the bilayer splitting exists in the non-CDW state and decreases as the interlayer separation increases. This shows that the coupling between the Te bilayers is the mechanism behind the bilayer splitting.

To further substantiate the above arguments, in R1:Fig. S9b we have calculated the band structure of LaTe_2 that has one La-Te block and a single Te layer, the structure is shown in R1:Fig. S9a. This resembles the top half of the non-CDW LaTe_3 structure i.e., the block above the blue dashed line in R1:Fig.S6b. For LaTe_2 in R1:Fig. S9b, we find two bands around $k_z=0$ that are the inner and the outer main bands arising from the Te p states in the single layer. Any splitting of these bands is not observed since the bilayer is absent in this structure. A similar comparison has been performed by Laverock et. al. (PRB **71**, 085114, 2005) between LuTe_3 and LuTe_2 to show the absence of the bilayer splitting in the latter. Here, we would like to add further that bilayer splitting has been related to the coupling between the Te bilayers in the literature of rare earth tellurides [Laverock et. al. PRB **71**, 085114 (2005), R1:Refs. 36, 50, 51], and our new calculations reaffirm that in the case of LaTe_3 .

Below we show that the other possible reasons for bilayer splitting suggested by the Reviewer are not supported by evidence. For example, there is no signature of “*surface reconstruction*” in LaTe_3 from our LEED and STM studies (see for example Figs. 1b,c, and Fig. S4). Moreover, the excellent agreement of ARPES with DFT for the bulk structure shows the absence of surface reconstruction. Furthermore, surface relaxation that can be considered as a “*genuine termination effect*” also may not be significant because the value of q_{CDW} determined from the three surface-sensitive techniques (LEED, STM, and ARPES) is close to the value from x-ray crystallography that is bulk-sensitive. Another argument that surface effects do not play a significant role is that our DFT calculations representing the bulk electronic structure of LaTe_3 (Fig. S15) agree fairly well with the experimentally determined bilayer splitting values (see Table: S2).

To answer the question “*What is the exact relationship between the CDW state and bilayer-splitting?*” we would like to point out that the bilayer splitting is observed in the band structure calculation for both the non-CDW as well as the CDW states of LaTe_3 . Note further that despite the absence of bilayer splitting in LaTe_2 discussed above, it hosts a CDW [Garcia et. al. PRL **98**, 166403 (2007), Shin et. al. PRB **72**, 085132 (2005)]. These reasonings show that the bilayer splitting in rare-earth tellurides could occur in both CDW as well as non-CDW states. In the CDW state, we have observed the bilayer splitting as shown in Figs. 2h-k in the EBS, and also in the corresponding folded band structure in R1:Fig. S15a-d. Quantitatively, the extent of the bilayer splitting in the CDW state is not expected to be very different from the non-CDW state, since the CDW distortion is only about 4% compared to the Te-Te distances. The bilayer splitting is nearly identical in both states, as shown in R1:Fig. S8e, where the bilayer splitting for the CDW state is obtained from R1:Fig. S15a. The above discussion has been included as a new Supplementary Note (#3) in the R1 version. Moreover, a part of this discussion is included in the 2nd paragraph of R1:page 9.

We hope that the above discussions also justify the title of the paper “*Charge density wave induced nodal lines in LaTe₃*”. It was decided after extensive deliberation between the co-authors because the interaction of the shadow bands with the main bands gives rise to the nodal lines, where the former are the manifestation of the CDW state.

Comment: 2. *Considering the recent ARPES results of NdTe₃ (arXiv: 2209.04226), especially, Fig.2 (b) inside, where the bilayer splitting is not observed, a natural question can be raised, say, what is the situation for LaTe₃? I recommend the authors add such details at least in the Supplementary information.*

Reply: Bilayer splitting has been observed in LaTe₃ in both the Fermi surface (see Fig. 1e and R1:Fig.S7) as well as in the bands from ARPES. It is also observed from theory which is in good agreement with ARPES, compare for example Figs. 2a-d with Figs. 2h-k.

The bilayer splitting is also observed in the ARPES data of NdTe₃ reported recently in R1:Ref. 63 [arXiv:2209.04226, presently published as Phys. Rev. B 107, L161103 (2023)]. This is evident from the statement in that paper: “We obtain ARPES results revealing not only the detailed fine structure of band interactions and bilayer splitting but also FS elements inside the CDW gap”. Moreover, in Figs. 2c and d of R1:Ref.63, the bilayer split bands are shown by the red arrows, and the authors further state: “The bilayer splitting can be detected for higher binding energies in both samples.” Bilayer splitting has been reported in other RTe₃ systems also such as Laverock et. al. PRB 71, 085114 (2005) and R1:Refs. 36, 50, 51.

Comment: 3. *Although overall the theoretical calculations match the ARPES data, I struggle to distinguish the bilayer splitting of the shadow bands in Fig.2 (a) and assume that the splitting tendency is small near E_F, while larger at around 0.8 eV, which is in contrast to the calculations in Fig.2 (h). Besides, the splitting features near E_F haven't been well resolved in the data. Since these band features seem to be quite close to E_F both experimentally and theoretically, have the authors tried potassium surface doping of the system to gain the features above E_F? And are there any attempts to produce few-layer material? This may also help us understand the mechanism behind the bilayer-splitting.*

Reply: We appreciate the Reviewer's comment that “*overall the theoretical calculations match the ARPES data*”. The bilayer splitting of the shadow band is also observed in Fig. 2a. To show this explicitly, in the R1 version, a zoomed view of the curvature plot in the near E_F region is shown in modified Fig. 2e (instead of the raw intensity plot in the R0 version). In Fig. 2e the bilayer splitting is indicated by a pair of red arrows for both the main and shadow bands. It is also observed near E_F on the negative k_z side, as shown in the Fig.A1 of our Reply.

Fig A1: (a) A zoomed portion near E_F of Fig. 2a of the main text and the MDC intensity profile with the fitted spectrum. The bilayer splitting of the outer band is established from the existence of a two-peak feature [fitted with two Lorentzian peaks (yellow peaks)] of the MDC profile at E_F . (b) A curvature plot of Fig. 2a shows the bilayer splitting of the outer bands in both the positive and negative k_z sides.

We agree with the Reviewer's observation that the tendency of splitting is smaller near E_F and increases with E . The bilayer splitting shows dependence on both E and k_x . The dependence of bilayer splitting on k is also observed for the γ pocket of the Fermi surface (see Supplementary Note 2).

To quantify the variation of the bilayer splitting (Δk_z) in the R1 version, we determine it at different E and k_x by fitting the MDCs from the ARPES intensity plots in Figs.2a-d with Lorentzian functions (as shown in Fig. A1a). This is shown in Supplementary R1:Table S2, which also contains the corresponding DFT values from Figs.2h-k for the 7f structure. In Fig.2a at $k_x = 0.68 \text{ \AA}^{-1}$, the Reviewer correctly observed that Δk_z increases with E . In addition, Table S2 reveals this trend for all k_x values (0.68 to 0.56 \AA^{-1} , i.e. Figs.2a-d). Increase of Δk_z with E is also clearly observed from DFT for k_x values ranging from 0.56 to 0.63 \AA^{-1} in Figs. 2k,j,i, whereas it is somewhat less prominent at $k_x = 0.68$ in Fig. 2h. Note that such dependence of bilayer-splitting on E was recently reported from an ARPES study of NdTe_3 ; see Fig.2d of R1:Ref.63. For LaTe_3 , we find this trend in similar E range as Ref. 63, where the bilayer split bands are of Te p_x - p_z character. However, for higher E , the out-of-plane Te p_y contribution is dominant (as discussed in R1:page 10 and illustrated in Supplemental Fig.S17), and as a result, the bilayer split bands are significantly modified. For example in Figs. 2c,d,j,k around $E = 0.7$ - 0.8 eV, the k_z separation and the slope of the bands change significantly. In Fig.2h, the bilayer split bands are also modified, but here these come closer at $E = 0.85$ eV. This is also observed in ARPES at $E = 0.95$ eV (see Fig. A2) showing that the agreement between ARPES and DFT at $k_x = 0.68 \text{ \AA}^{-1}$ is reasonable.

Fig A2: A curvature plot of Fig. 2a, the green dashed curves serve as a guide to the eye. The red horizontal arrows show the modification of bilayer split bands at $E = 0.95$ eV.

It is interesting to note that even for the non-CDW state, our DFT calculation in Fig. S8a at $k_x = 0.68 \text{ \AA}^{-1}$ shows the same trend of increase in Δk_z . Furthermore, the Δk_z values shown in the rightmost column of Table S2 are nearly alike between the non-CDW and the CDW states. Moreover, the modification of bands at higher E , such as the proximity of the bilayer split bands around 0.85 eV is also observed (Fig. S8a). Thus, the above discussed behavior of the bilayer split bands seems to be related to the details of the band structure of undistorted LaTe_3 . A part of the above discussion is included in the main text in the 1st paragraph of R1: page 11.

In what follows, we would like to dwell on the other comments. The Reviewer has asked “*have the authors tried potassium surface doping of the system to gain the features above E_F ?*” in the context of the origin of the bilayer splitting.

Deposition of K is a standard method in photoemission to study the bands above the Fermi level assuming that there is a rigid band shift due to band filling by the electrons donated by the electropositive alkali metal. However, since the CDW in LaTe_3 is sensitive to the Te band filling there is a possibility that q_{CDW} could change, which would lead to a change in the band structure itself rather than a rigid band shift. The Reviewer’s question made us curious and we performed new experiments on LaTe_3 with K deposition. The surprising finding is that even with the smallest deposition (30 sec) at room temperature q_{CDW} increases. This would modify the bands, as the shadow band is shifted from the main band by q_{CDW} . This rules out the rigid band shift model that is required for studying bands above E_F with K deposition. The other interesting observation with larger deposition is the appearance of a K related surface reconstruction that completely changes the ARPES bands. The change in q_{CDW} with K deposition time and the corresponding LEED patterns are shown in Fig.A3.

Fig A3: q_{CDW} as a function of K deposition time at room temperature. The CDW superstructure spots are indicated by the green arrows in the LEED images measured with 40 eV primary electron beam energy. The blue arrows in 390 and 690 sec depositions show the new spots related to a K related surface reconstruction.

The experiments were also performed by depositing K at a substrate temperature of 120 K, and the results were largely similar. We thank the Reviewer for suggesting this work that we will further extend in the future.

In the context of the bilayer splitting, it was asked “*And are there any attempts to produce few-layer material?*”

Few as well as single layer transition metal dichalcogenides (TMDCs) have been studied extensively in the literature [Chen et al., *Nature Communications* **6**, 8943 (2015), Sugawara et al., *ACS Nano* 2016, 10, 1, 1341–1345, Zhao et al., *Nature Nanotech* **10**, 765–769 (2015), Ugeda et al., *Nature Phys* **12**, 92–97 (2016) and Sanders et al., *PRB* 94, 081404(R) (2016)], but there is hardly any such report for LaTe_3 . The studies on TMDCs convey that in addition to the challenges in the preparation and characterization of monolayers, the substrate might play an important role in the ultrathin limit. For example, Sanders et al. showed that single-layer 1H-TaS_2 grown epitaxially on $\text{Au}(111)$ does not undergo a CDW transition down to $T = 4.7$ K, whereas bulk 2H-TaS_2 develops a 3×3 CDW periodicity below about 75 K. This was explained by n -doping of the material due to the substrate. In contrast, for the 1H-TaSe_2 monolayer on $6\text{H-SiC}(0001)$ the CDW state is observed (Ryu et al., *Nano Lett.* 18, 689, 2018). Thus, a study of the intrinsic properties of few-layer LaTe_3 should be done on a substrate that would negligibly affect its CDW related properties. However, this may not be straightforward since from our preliminary study of K/LaTe_3 (see previous paragraph), we find that electron doping modifies the CDW state. Moreover, epitaxial growth of few layers of LaTe_3 on a lattice matched substrate using molecular beam epitaxy (as preparation by exfoliation in the air would not work since the material is air sensitive) and its characterization would be an interesting study by itself. MBE growth of LaTe_3 is not reported in the literature. The bilayer splitting is

expected to be observed since Te bilayers are part of the unit cell, although the magnitude of the splitting might change depending on the substrate. But again, this can be an entirely new study as the aim of the present paper is to study the band structure of LaTe₃ crystal. We hope that our present work will motivate further research in this direction.

Comment: 4. *The nodal line related data presented in the manuscript is abstract and sometimes hard to follow. I understand there is energy dispersion along the k_x direction for the nodal lines, still it would be relatively easy for us to understand if the authors present the energy contours of the k_x - k_z plane.*

Reply: In the R1 version, the k_x - k_z isosurface plots at different E ranging from 0 to 0.6 eV at a step of 0.1 eV are shown in R1:Fig. S13. The curvature plot in R0:Fig. S8 is zoomed from the $E=0.5$ eV isosurface. The sentence in the 3rd paragraph of R1:page 9 is modified accordingly.

Comment: 5. *The authors do mention the orbital characteristics of the bands and show the p -orbital characters from the EBS calculations in Fig. S12. However, some essential information on the ARPES experiment is missing. What is the photon polarization used, and what is the measurement scheme of the polarization-dependent ARPES? I recommend the authors add the information wherever appropriate.*

Reply: We have added this information about the ARPES experiment in the Methods section. A schematic diagram of the experimental geometry is also shown in the Supplementary information as Fig. S26.

Comment: 6. *The ab initio DFT calculation is based on a realistic experiment-based structure. Usually, the band structure topology is very sensitive to the crystal constants, have the authors performed calculations using just the theoretical self-consistent crystal structure? These calculated results should be added to the Supplementary information as reference.*

Reply: Following the Reviewer's suggestion, we have carried out DFT calculations with complete geometry optimization for the 7f structure including the van der Waals interaction term (DFT-D3 method). However, the electron-phonon interaction was not considered. The CDW related modulation is observed after optimization and the q_{CDW} value remains practically unchanged (0.2855c*). There is a marginal decrease in the lattice constants by 0.4-0.6%. But, the CDW amplitude decreases drastically by 80% compared to the experimental value from x-ray crystallography. Also, the CDW gap along ΓZ is not obtained, as shown in Supplementary Fig. S27a. The EBS calculation shows that the spectral weight of the shadow bands is much lower and so these are virtually non-existent (Supplementary Fig. S27b). These results from DFT with optimization disagree starkly with ARPES and DFT without optimization where the CDW gap along ΓZ (0.45 eV from ARPES and 0.35 eV from DFT without optimization, see Figs. 4, S11) and the shadow bands (Fig. 2) are clearly observed. Thus, the use of the experiment based structure without optimization for performing DFT is justified.

In fact, the use of the experimental structure for DFT without optimization is a prevalent practice in the scientific literature, particularly for large unit cell structures with modulation and/or anti-site defects. See for example our recent work Bhattacharya *et al.* arXiv:2304.04992v1 (2023) [Ref R1:80], as well as Refs. R1:27,78,79, to mention a few.

A part of this discussion is included in the Methods section in the 2nd paragraph of page 21.

Comment: *Additional comments: 7. Such as the bottom axis of Fig.3 (m), which is not consistent with Fig.3 (l), the authors should correct these inconsistencies since the band crossings should locate at the same kz point.*

Reply: We have rectified this problem in Fig.3 in the R1 version.

Comment: *8. Also in the caption of Fig.3, the authors mentioned that “The vertical dashed line represents the kz point on Γ_2X_2 i.e., the Σ line.” I checked that in Fig.1 (d) where the “ Σ ” label is with the gray arrow pointing at the Γ -X line. Is this a typo? A highlighted guiding line connecting the high-symmetry points in 3D would be also very helpful.*

Reply: We thank the Reviewer for pointing out this. Γ_2X_2 is the Σ line in the 2nd BZ, this is now mentioned in the caption of Fig.3. The 3D BZ showing the important direction is provided in Fig. 1d. This figure is modified to show the Γ_2X_2 direction by green color. Note that all the high symmetry directions and points of the CDW BZ are shown in Fig. S6c, and this is also referred to in the caption of Fig. 1d in the R1 version.

Comment: *In conclusion, this manuscript deserves further consideration. I would suggest the authors to do this sort of study and revision before resubmission as it is crucial for interpreting both the ARPES and calculated results.*

Reply: We have performed a major reworking of the R1 version of the manuscript. We have responded to all queries and addressed the reviewer's suggestions by performing new DFT calculations and conducting various new experiments. We think this exercise has made a big difference in the quality of our manuscript and we are grateful to the Reviewer for that.

(please turn over)

Summary of the Major changes in R1 version

- 1) *DFT calculations in Section III, R1: Figs. 5, S24, S25.*
We have performed fresh DFT band structure calculations for a large unit cell (29-fold) of LaTe_3 comprising 232 atoms. These calculations reaffirm the existence of KNL state in LaTe_3 and other significant findings from the calculations performed for the 7-fold structure.
- 2) *DFT calculations in R1: Figs.S8 and S9.*
 - a) New DFT results of the non-CDW LaTe_3 for three different structures where the Te-Te bilayer distance has been increased by 0%, 5% and 10% to explain the mechanism of the bilayer splitting in LaTe_3 .
 - b) We have included DFT band structure data for LaTe_2 containing a single Te layer to show the absence of bilayer splitting.
- 3) *EBS calculations in R1: Fig.S18.*
We have compared folded band structure and unfolded EBS in the crossing region calculated w/o SOC and with SOC.
- 4) *DFT calculations in R1: Fig.S27.*
New DFT results for the relaxed 7-fold CDW structure. It shows the non-existence of the CDW gap along ΓZ and the shadow bands near the crossing region. This justifies the use of the experiment based unrelaxed structure.
- 5) *Experimental data in R1: Fig. S5.*
New Raman spectroscopy data measured in the CDW phase of LaTe_3 to establish its noncentrosymmetry.
- 6) *Experimental data in Reply to Reviewer #3 (Fig. A3)*
LEED data for K/LaTe_3 as a function of deposition time showing the change in q_{CDW} .
- 7) *Analysis of the experimental data:*
 - a) 2d curvature analysis of R0: Fig. 2e and shown in R1: Fig.2e to establish the existence of the bilayer splitting in both the shadow and main outer bands.
 - b) A supplementary table (Table S2) to compare the bilayer splitting observed from ARPES and DFT for the outer main bands in the CDW phase.
 - c) A series of k_x - k_z isosurface plots are now shown in stacked form to show the crossings on the isosurfaces.
 - d) Recalculated the error in the q_{CDW} value from LEED intensity line profiles.
 - e) We have provided new supporting evidence of the inversion symmetry breaking from our STM result shown in R1: Fig.1b, as well as Fig. S3
- 8) *Presentation:*
 - a) In Fig. 1d, both the ΓX and $\Gamma_2 X_2$ lines (Σ lines) are highlighted by green lines. These lines are now consistent with the KNL shown in green colour in Fig. 3o.

- b) We have modified the illustrations shown in R1: Fig. 3h,o to show the energy dispersion of the nodal lines.
- c) The x-scales in all the three panels (a-c) of Fig. 4 are now consistent with each other.
- d) We have added a new figure R1: Fig. S26 to demonstrate the experimental geometry of the ARPES measurements.
- e) Improved the visibility of the yellow and blue markers in R1: Fig. 3f,g.
- f) All figures in the main text and supplementary information are now provided in high resolution PDF format (earlier were in '.png') for better clarity of the data.
- g) To improve the visualization of the effect of CDW modulation on the Te net, we have removed R0: Fig. S1 and included two new figures R1: Fig. S2 and R1: Fig. S3.
- 9) *New references added in R1 version:*
37, 41-45, 54,55,73,78-81,84-87

Summary of the changes in the figures in the R1 version

Main text figures		
Revised submission (R1)	Original submission (R0)	Summary of changes
Fig. 1a,c,e	Fig. 1a,c,e	No change.
Fig. 1b	Fig. 1b	Minor change
Fig. 1d	Fig. 1d	Minor change
Fig. 2a-d, g	Fig. 2a-d, g	No change.
Fig. 2e	Fig. 2e	Replaced the raw ARPES figure with its 2d curvature plot.
Fig. 2f	Fig. 2f	Minor change
Fig. 2h-k	Fig. 2h-k	Minor change
Fig. 3a-e, i-n	Fig. 3a-e, i-n	No change.
Fig. 3f,g	Fig. 3f,g	Minor change
Fig. 3h,o	Fig. 3h,o	Replaced the previous images with new illustrations to explicitly show the energy dispersion of the nodal lines.
Fig. 4a-c	Fig. 4a-c	Minor change
Fig. 5		A new figure showing the DFT results for the 29-fold LaTe ₃ .

Supplementary figures and tables		
Revised submission (R1)	Original submission (R0)	Summary of changes
Fig. S1		New figure: shows 29-fold structure of LaTe ₃ .
Fig. S2		New figure: comparison of Te atom positions of incommensurate, 29-fold and 7-fold structures of LaTe ₃ .
Fig. S3		New figure: shows how noncentrosymmetry and polar direction arise in the CDW state.
Fig. S4	Fig. S2	No change.

Revised submission (R1)	Original submission (R0)	Summary of changes
Fig. S5		New figure: Identification of noncentrosymmetric eigenmodes from Raman data.
Fig. S6a,b	Fig. S3a,b	Minor change
Fig. S6c	Fig. S6c	No change.
Fig. S7	Fig. S4	No change.
Fig. S8		New figure: explanation of the origin of bilayer splitting in LaTe ₃ .
Fig. S9		New figure: DFT band structure of LaTe ₂ , discussed in the 'Supplementary note 3'.
Fig. S10	Fig. S5	No change.
Fig. S11	Fig. S6	No change.
Fig. S12	Fig. S7	Added a new panel b showing the non-CDW structure of LaTe ₃ .
Fig. S13a	Fig. S8a	Shows k _x -k _z isosurface plots measured at seven different E, R0 version had one isosurface plot.
Fig. S13b	Fig. S8b	No change.
Fig. S14	Fig. S9	No change.
Fig. S15	Fig. S10	No change.
Fig. S16a	Fig. S11a	Replaced DFT calculated at k _x =0.72 Å ⁻¹ in R0 version with a calculation performed at k _x = 0.68 Å ⁻¹ for better comparison with ARPES (panel b).
Fig. S16b	Fig. S11b	No change.
Fig. S17	Fig. S12	No change.
Fig. S18		New figure: shows the comparison of folded band structure and EBS for the 7-fold structure.
Fig. S19	Fig. S13	No change.
Fig. S20	Fig. S14	No change.
Fig. S21	Fig. S15	No change.
Fig. S22	Fig. S16	Minor change
Fig. S23	Fig. S17	No change.
Fig. S24		New figure: New DFT band structure results for the 29-fold LaTe ₃ establishing the crossings.
Fig. S25		New figure: Evidence of KNL from new DFT calculation for 29-fold LaTe ₃ .
Fig. S26		New figure: Illustration of the experimental geometry of the ARPES measurements.
Fig. S27		New figure: DFT results of the relaxed 7-fold structure of LaTe ₃ .
Fig. S28	Fig. S18	Minor change in caption.
Table S1	Table S1	Minor change in caption.
Table S2		New table: comparison of bilayer splitting between ARPES and DFT.
Table S3	Table S2	No change.
Table S4	Table S3	No change.

Reviewers' Comments:

Reviewer #1:

Remarks to the Author:

My comments are properly addressed by the authors. I suggest the publication of this manuscript.

Reviewer #2:

Remarks to the Author:

The authors have provided detailed replies and additional data to address all my comments and concerns. This manuscript should be published without further delay.

Reviewer #3:

Remarks to the Author:

The manuscript "Charge density wave induced nodal lines in LaTe_3 " by Shuvam Sarkar et al. has certainly improved since the last time and it is obvious that the authors have put efforts in addressing the comments previously made.

The authors have addressed the concerns regarding the relationship between the CDW state and bilayer-splitting, band splitting tendency, and experiments of potassium surface doping, etc. Particularly, the discussion about the relationship between the CDW state and bilayer-splitting is now clear and reasonable. The new DFT calculations added to the revised manuscript for the reasons behind the bilayer-splitting is now straightforward and well presented, this is also for the DFT calculation based on the realistic experiment-based structure. Potassium surface doping experiments have been properly performed that rules out the rigid band shift model.

The authors also incorporated the comments into the revised manuscript appropriately.

Based on the nice work done by the authors I recommend this manuscript for publication.

REPLY TO THE REVIEWER COMMENTS

Reply to the comments of Reviewer #1:

Comments: *My comments are properly addressed by the authors. I suggest the publication of this manuscript.*

Reply: We appreciate the reviewer's recommendation to publish our manuscript in Nature Communications.

Reply to the comments of Reviewer #2:

Comment: *The authors have provided detailed replies and additional data to address all my comments and concerns. This manuscript should be published without further delay.*

Reply: We thank the reviewer for recommending that our work be published as soon as possible.

Reply to the comments of Reviewer #3:

Comment: *The manuscript "Charge density wave induced nodal lines in LaTe₃" by Shuvam Sarkar et al. has certainly improved since the last time and it is obvious that the authors have put efforts in addressing the comments previously made.*

The authors have addressed the concerns regarding the relationship between the CDW state and bilayer-splitting, band splitting tendency, and experiments of potassium surface doping, etc. Particularly, the discussion about the relationship between the CDW state and bilayer-splitting is now clear and reasonable. The new DFT calculations added to the revised manuscript for the reasons behind the bilayer-splitting is now straightforward and well presented, this is also for the DFT calculation based on the realistic experiment-based structure. Potassium surface doping experiments have been properly performed that rules out the rigid band shift model.

The authors also incorporated the comments into the revised manuscript appropriately. Based on the nice work done by the authors I recommend this manuscript for publication.

Reply: We appreciate the reviewer's compliment and recommendation for publication.